# Cyclodipeptide oxidase is an enzyme filament

Michael P. Andreas[1] & Tobias W. Giessen ●[1] ✉

Modified cyclic dipeptides represent a widespread class of secondary metabolites with diverse pharmacological activities, including antibacterial, antifungal, and antitumor. Here, we report the structural characterization of the *Streptomyces noursei* enzyme AlbAB, a cyclodipeptide oxidase (CDO) carrying out α,β-dehydrogenations during the biosynthesis of the antibiotic albonoursin. We show that AlbAB is a megadalton heterooligomeric enzyme filament containing covalently bound flavin mononucleotide cofactors. We highlight that AlbAB filaments consist of alternating dimers of AlbA and AlbB and that enzyme activity is crucially dependent on filament formation. We show that AlbA-AlbB interactions are highly conserved suggesting that other CDO-like enzymes are likely enzyme filaments. As CDOs have been employed in the structural diversification of cyclic dipeptides, our results will be useful for future applications of CDOs in biocatalysis and chemoenzymatic synthesis.

Cyclic dipeptides and their derivatives represent an important class of secondary metabolites[1–4] with diverse pharmacological activities ranging from antibacterial and antifungal to antitumor and antiplasmodial[5–11]. Cyclic dipeptides are involved in bacterial quorum sensing[12,13] and show high cell penetration proficiencies and protease resistance compared to acyclic peptides[2]. Structurally, cyclic dipeptides are defined by their heterocyclic 2,5-diketopiperazine (DKP) backbone formed by the condensation of two α-amino acids. DKP formation can be catalyzed by three evolutionarily unrelated types of enzymes: nonribosomal peptide synthetases (NRPSs)[14–21], cyclodipeptide synthases (CDPSs)[22–28], and arginine-containing cyclodipeptide synthases (RCDPSs)[29]. While DKP-forming NRPSs are primarily found in fungi, the vast majority of CDPSs are encoded in bacterial genomes. The only recently identified RCDPSs seem to be confined to the fungal kingdom[29]. NRPSs use free α-amino acids as substrates for cyclic dipeptide formation. In contrast, CDPSs and RCDPSs utilize aminoacyl-tRNAs as substrates[30–35], diverting them from their canonical role in ribosomal protein synthesis.

Cyclic dipeptide formation is generally followed by further modification of the DKP backbone through dedicated and co-regulated tailoring enzymes[24,25,36]. For example, CDPS-dependent pathways encode a wide variety of DKP-modifying enzymes including cytochrome P450s, Fe[II]/2-oxoglutarate-dependent oxygenases, prenyltransferases, methyltransferases, terpene cyclases, and cyclodipeptide oxidases (CDOs)[36–42]. As CDPSs show broad substrate specificities, substantial effort has been dedicated towards utilizing them in combination with promiscuous DKP tailoring enzymes for the combinatorial biosynthesis of novel modified DKPs with modulated or improved bioactivities[38,39,43–45]. For example, CDOs have been used for the diversification of DKP scaffolds through α,β-dehydrogenations[46,47]. To date, the CDO-containing guanitrypmycin[48], purincyclamide[49], nocazine[38], and albonoursin[50,51] pathways have been the focus of biosynthetic studies. Beyond the original partial isolation of the albonoursin CDO from its native host *Streptomyces noursei* – which yielded valuable foundational information about CDO function – CDOs have so far only been investigated or utilized in vivo or in crude extracts[46,47]. This is due to difficulties in heterologously producing and purifying CDOs which has also prevented their structural analysis.

Here, we report the structural characterization of the *S. noursei* CDO – AlbAB – involved in albonoursin biosynthesis. We show that AlbAB is a megadalton heterooligomeric enzyme filament containing covalently bound flavin mononucleotide (FMN) cofactors. We further show that filaments consist of alternating dimers of AlbA and AlbB and that enzyme activity is crucially dependent on filament formation. We highlight that AlbA-AlbB interactions within the filament are highly conserved suggesting that all CDOs are likely enzyme filaments. We substantially expand the number of known CDO-like enzymes and show that the majority of them are found outside of CDPS gene

[1]Department of Biological Chemistry, University of Michigan Medical School, Ann Arbor, MI 48109, USA. ✉e-mail: tgiessen@umich.edu

clusters. This study reports the structural characterization of a CDO, explains the historically encountered difficulties in working with CDOs, and provides molecular level detail about CDO structure and function. These insights will be useful for future biotechnological applications of CDOs as enzyme catalysts in biocatalysis and chemoenzymatic synthesis.

## Results

### Distribution and diversity of CDO-containing gene clusters

To date, ten CDOs have been experimentally studied, mostly with respect to their ability to carry out α,β-dehydrogenations in various cyclic dipeptides, primarily for structural diversification and bioproduction purposes[38,48–51]. The biosynthetic roles for only four CDOs – AlbAB (albonoursin)[50,51], Ndas_1146/1147 (nocazines)[38], GutBC (guanitrypmycins)[48], and PcmBC (purincyclamides)[49] – have been elucidated so far (Fig. 1a, b). All previously discovered CDOs consist of

two components – designated CDOA and CDOB – encoded by two separate, often overlapping, genes with strong gene synteny. CDOAs are predicted to share sequence and structural similarity with nitroreductases (PF00881), a large and diverse superfamily of flavoproteins, while CDOBs exhibit no sequence or structural homology to any characterized protein family (PF19585). All so far investigated CDOs are found within biosynthetic CDPS gene clusters and carry out α,β-dehydrogenations on cyclic dipeptides assembled by tRNA-dependent CDPSs (Fig. 1c).

We computationally identified 274 CDO-containing gene clusters found across 18 bacterial and 9 archaeal phyla (Fig. 1d). CDO-encoding genes are most abundant in the phyla Actinomycetota, Bacillota, Chloroflexota, Euryarchaeota, and Thermoproteota. Analysis of the identified gene clusters reveals diverse CDO genome neighborhoods with only a minority of them (12%) encoding CDPS genes (Supplementary Fig. 1). Consistent with previous bioinformatic analyses[46], all

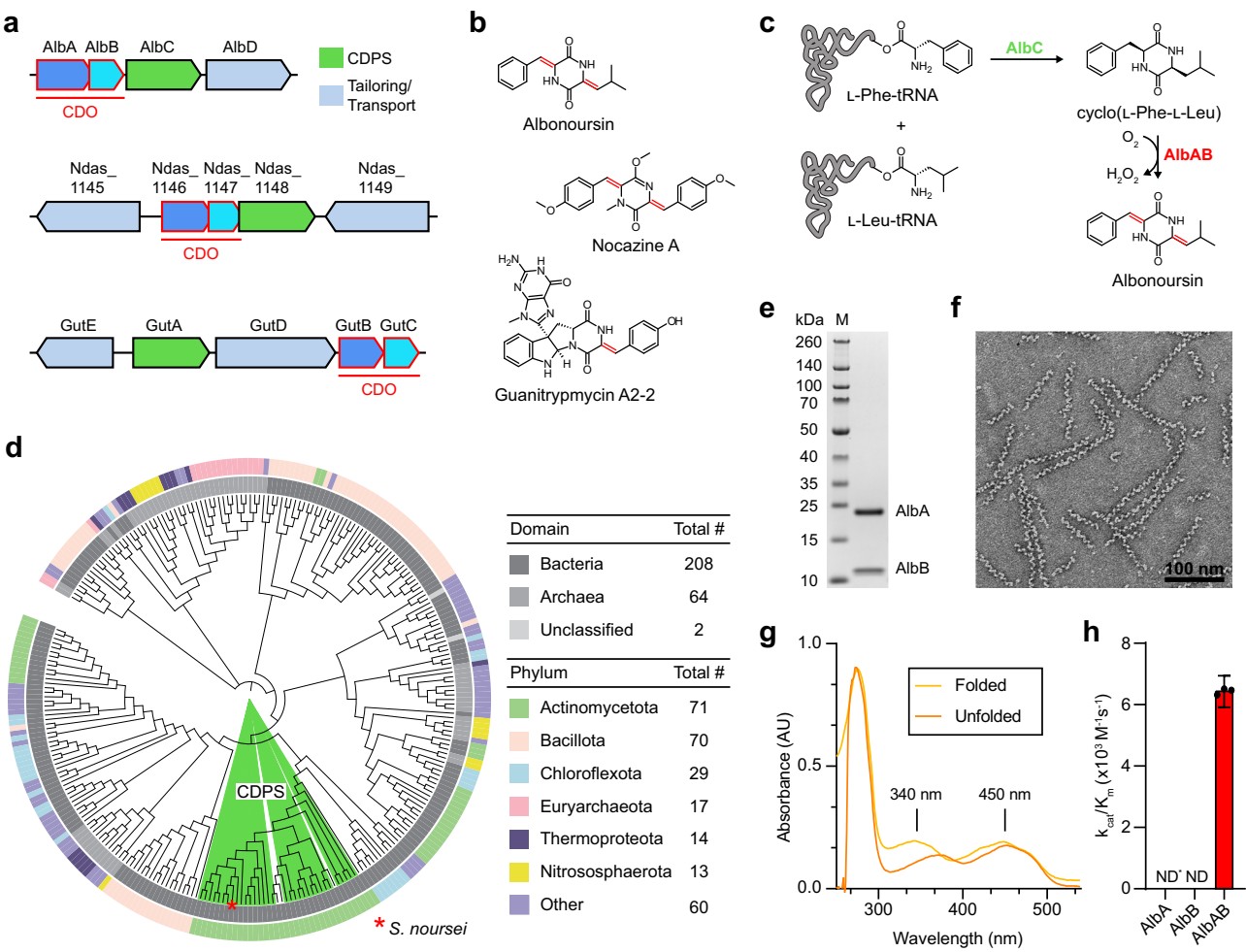

**Fig. 1 | Diversity and phylogenetic distribution of CDOs and characterization of filamentous AlbAB. a** Three characterized CDPS and CDO-encoding gene clusters. **b** Natural products produced by the gene clusters shown in (**a**). Double bonds formed by CDO-catalyzed α,β-dehydrogenations are highlighted (red). **c** Biosynthesis of albonoursin. AlbC catalyzes a double condensation reaction using L-Phe-tRNA and L-Leu-tRNA to produce cyclo(L-Phe-L-Leu). The CDO AlbAB subsequently catalyzes the dehydrogenations of the two Cα-Cβ bonds, transferring the electrons to oxygen (O₂) to yield hydrogen peroxide (H₂O₂). **d** Phylogenetic tree of 274 CDOs. CDOs found in CDPS-containing gene clusters are highlighted (green). The outer ring coloring represents the phyla distribution and the inner ring coloring represents the domain distribution. A table detailing the phyla and domain distributions is shown (right). **e** SDS-PAGE analysis of purified AlbAB heterologously expressed in *Streptomyces coelicolor*. M: molecular weight marker. Source data are

provided as a Source Data file. This experiment was repeated independently three times. **f** Negative stain TEM micrograph of purified AlbAB highlighting filament formation. This experiment was repeated independently four times. **g** UV-Vis spectrum of folded and unfolded AlbAB. Both show characteristic flavin absorptions, indicating that FMN is covalently bound. Source data are provided as a Source Data file. **h** Enzymatic activity of purified filamentous AlbAB against the substrate cyclo(L-Phe-L-Leu). ND* not detected, AlbA could not be expressed in a soluble form by itself. ND Not detected, AlbB could be purified by itself but exhibited no measurable activity against cyclo(L-Phe- L-Leu). $k_{cat}/K_{m}$: catalytic efficiency. Data are shown as mean values with error bars representing standard deviation of three independent experiments. Source data are provided as a Source Data file.

33 identified CDPS-containing gene clusters are present in the phylum Actinomycetota, with 70% of them found in the single genus *Streptomyces*. The majority of identified CDO-containing operons (88%) do not contain CDPS genes, with 18 of them representing previously unidentified fusion systems where the two CDO components – CDOA and CDOB – are fused into a single polypeptide chain connected by a long flexible linker (Supplementary Fig. 1). Examination of non-CDPS-encoding gene clusters reveals the presence of many genes associated with hydantoin or allantoin modification, suggesting a role in purine metabolism[52,53] or the biosynthesis of novel peptide-based secondary metabolites[54]. In particular, genes coding for proteins with homology to hydantoin racemases (PF01177), DUF917 family proteins (PF06032), and AroM family proteins (PF07302) are enriched in these gene clusters. While DUF917 family proteins have not been experimentally characterized, sequence and structure comparisons suggest them to be hydantoinases able to hydrolyze hydantoin-like heterocyclic rings to the corresponding *N*-carbamoylated linearized products[55,56]. AroM family proteins have similarly not been experimentally characterized, however, their predicted structures are nearly identical to hydantoin and glutamate/aspartate racemases, suggesting that they likely possess isomerization activity[57,58]. Genes encoding for transport-associated proteins are also enriched in CDO-encoding gene clusters. Examples include ATP-binding cassette (ABC) dipeptide/oligopeptide transporters[59–61], OPT family transporters (PF03619)[62], and EamA domain-containing proteins (PF00892)[63–67]. Further, different types of peptidase genes are found in many non-CDPS CDO-encoding gene clusters and may function to process exogenous or endogenous peptides that could serve as substrates for other operon components, including CDO-like enzymes[58].

Notably, we also identified CDOs in anaerobic bacteria and archaea belonging to the genera *Blautia*, *Clostridium*, and *Thermococcus*. As all so far characterized CDOs rely on oxygen to re-oxidize their FMN cofactors, CDOs in anaerobes may utilize different terminal electron acceptors.

## AlbAB is a filament-forming CDO with covalently bound flavin cofactors

AlbAB was heterologously expressed and purified using *Streptomyces coelicolor* as the expression host. Analysis of the purified sample by SDS-PAGE clearly showed the presence of two proteins, AlbA at ca. 21 kDa and AlbB at ca. 11 kDa in approximately equimolar amounts (Fig. 1e). As reported previously[50], we observed the majority of AlbAB eluting at or near the void fractions when purified using a Superose 6 Increase 10/300 GL column, suggesting that the majority of AlbAB oligomers possess apparent molecular weights exceeding 2 megadaltons. Subsequent negative stain transmission electron microscopy (TEM) analysis revealed that AlbAB appears to form linear filaments of varied length with the majority of observed filaments between 100 and 300 nm in length. Filaments are approximately 10 nm wide and exhibit an estimated pitch of ~13 nm (Fig. 1f). AlbAB filaments were stable between pH 5.5 and 8.5 and from 0 to 1 M NaCl (Supplementary Fig. 2a). We further sought to identify a pH condition that disrupted filament assembly. We tested filament stability at pH 4.1, which corresponds to the pI of AlbB, and did not observe filaments suggesting that filament assembly is disrupted at low pH. Additionally, we observed a small fraction of filaments that appeared as intertwined or aggregated bundles, especially under low ionic strength conditions (Supplementary Fig. 2b). AlbAB is a flavoprotein with absorbance peak maxima at 343 nm and 448 nm, indicative of a bound flavin cofactor (Fig. 1g). These absorbance maxima are consistent with previously reported values for AlbAB of 343.5 nm and 447.5 nm[50]. We found that flavin absorbance was still present after AlbAB denaturation in 6 M guanidinium hydrochloride followed by washing steps, confirming that the flavin cofactor is covalently bound as previously reported[50]. Additionally, we observed a shift in the flavin absorbance maxima to

ca. 372 nm and 452 nm in denatured AlbAB. The flavin absorbance maxima in both folded and denatured AlbAB are consistent with other nitroreductases and flavin-binding proteins, most of which bind flavins non-covalently and generally exhibit absorbance maxima at ca. 360 nm and 450 nm, although some deviation in absorbance must be expected due to the local protein environment surrounding the flavin cofactor[68–71].

To test if purified filamentous AlbAB is enzymatically active, we carried out standard saturation kinetics analysis using the known preferred substrate cyclo(L-Phe-L-Leu) in an established coupled assay with horseradish peroxidase to detect hydrogen peroxide, stoichiometrically produced by AlbAB[50]. AlbAB showed substrate dependent enzymatic activity with a catalytic efficiency ($k_{cat}/K_M$) of $6.4 \times 10^3\,M^{-1}\,s^{-1}$, in good agreement with previously reported kinetic parameters (Fig. 1h and Supplementary Fig. 3b)[50]. We next set out to test the catalytic activity of the individual subunits. After testing both *S. coelicolor* and *E. coli* as production hosts, we were unable to express AlbA in a soluble form. AlbB on the other hand was soluble and could be purified (Supplementary Fig. 2a), however, it did not show any measurable enzymatic activity in the absence of AlbA (Fig. 1h). TEM analysis of reaction mixtures highlighted that AlbAB filaments remained assembled during enzyme assays whereas AlbB was incapable of forming filaments under assay or any other conditions (Supplementary Fig. 3c). The filamentous nature of AlbAB was further confirmed by size exclusion chromatography and dynamic light scattering experiments (Supplementary Fig. 3d,e). The same analyses suggested AlbB to form a dimer in solution (Supplementary Fig. 3d, e). Taken together, AlbAB behaves as an obligate heterooligomeric enzyme filament where filament formation is crucially necessary for catalytic activity and likely for solubilizing the flavin-bearing AlbA subunit.

## Single particle cryo-EM analysis of the AlbAB filament

To investigate the molecular structure and assembly of the AlbAB filament, we carried out single particle cryo-electron microscopy (cryo-EM) (Supplementary Fig. 4 and Supplementary Table 1). Initial 2D class averages displayed clear periodicity highlighting two distinct subunits along the length of the filament (Fig. 2a). 3D helical reconstruction yielded a map with a global resolution of 3.14 Å where a single $C_2$ symmetrical dimer of AlbA ($AlbA_2$) and a single $C_2$ symmetrical dimer of AlbB ($AlbB_2$) form a heterotetramer representing the biologically relevant asymmetric unit of the filament (Fig. 2b). The AlbAB filament lacked rotational symmetry along the long axis, instead exhibiting repeating units of $AlbA_2$ and $AlbB_2$ with a helical rise of 46 Å per $AlbA_2$ and $AlbB_2$, a helical twist of 120°, a helical pitch of 138 Å, and an overall diameter of 100 Å (Fig. 2c, d and Supplementary Movie 1). The identity of the previously noted flavin cofactor of AlbAB could be confirmed as flavin mononucleotide (FMN). One FMN cofactor per AlbA monomer could be unambiguously localized in our cryo-EM density. Masked local refinement containing one $AlbA_2$ and two surrounding $AlbB_2$ units yielded an improved 2.78 Å map allowing confident model building for the dimers $AlbA_2$ and $AlbB_2$ as well as the bound FMN cofactors (Fig. 2e–g and Supplementary Fig. 5 and 6).

AlbA adopts an α+β fold homologous to FMN- and NAD(P)H-dependent nitroreductase (NTR)-fold proteins[72]. The AlbA monomer consists of five α helices and four anti-parallel β strands (Supplementary Fig. 7a). AlbA dimer formation is mediated by extensive interactions between helix α4 of both monomers, which is further stabilized by interactions with helix α1. Strand β4 is part of a C-terminal extension which spans across the surface of both AlbA monomers and interacts with the anti-parallel β sheet formed by strands β1-β3 of the second monomer, a conformation almost always observed in NTR-like proteins[72]. The overall fold of AlbA is structurally similar to that of minimal, or "hub" NTRs[72]. NTRs represent a diverse family of proteins with three major sites of deviation from the minimal "hub" subfamily. AlbA only deviates from other "hub" NTRs by a slight elongation at the

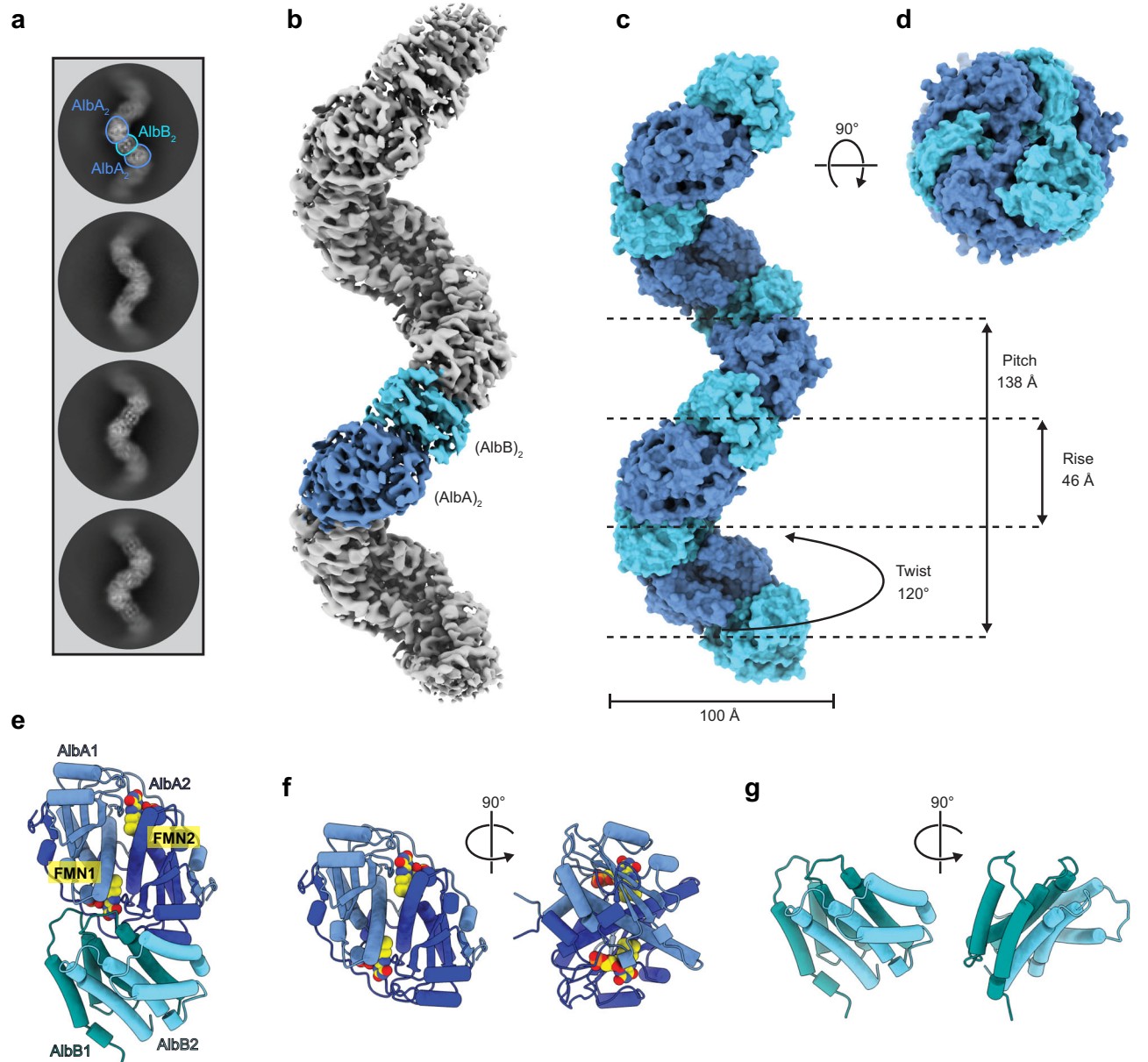

**Fig. 2 | Cryo-EM analysis of the AlbAB filament. a** Representative 2D class averages of filament particles, demonstrating two distinct subunits corresponding to AlbA$_2$ and AlbB$_2$ arranged in a periodic assembly. **b** Cryo-EM density resulting from a helical reconstruction of the AlbAB filament. AlbA$_2$ and AlbB$_2$ dimers are shown in dark blue and light blue, respectively. **c** Surface representation of the refined models for AlbA$_2$ and AlbB$_2$ assembled into a filament highlighting the geometric parameters of the helical filament. **d** View facing down the long axis of the filament highlighting its right-handedness. **e** Ribbon representation of the biologically relevant asymmetric unit of the AlbAB filament containing a dimer of AlbA and a dimer of AlbB in a heterotetrameric assembly. Covalently bound FMN cofactors are shown (yellow, spheres), highlighting their proximity to the AlbA$_2$-AlbB$_2$ interface. **f** Ribbon representation of a dimer of AlbA$_2$ with FMN shown (yellow, spheres). **g** Ribbon representation of the AlbB$_2$ dimer.

"E2" site between helix α4 and strand β2 (Supplementary Fig. 7b). A further distinguishing characteristic of AlbA is that residues 102-107 do not form a helix, which would correspond to helix α4 in most other NTRs. This feature has been observed in other NTR-like proteins, however, in AlbA it likely plays a role in the interaction with AlbB. Sequence comparison of AlbA with other characterized NTR-like proteins shows that AlbA is most closely related to NTRs involved in FMN fragmentation, dehalogenation reactions, and FMN modification[73–75] (Supplementary Fig. 7c).

The structure of AlbB presented here establishes the fold of CDOB proteins. AlbB forms a dimeric 8-helix bundle with each monomer containing four α helices arranged in an anti-parallel orientation similar to four-helix bundle proteins (Fig. 2e–g and Supplementary

Fig. 8a). The AlbB dimer is formed through extensive intermolecular interactions of 35 residues contained within helices α1, α2, and α4 in a knobs-into-holes packing scheme often found in four-helix bundle or coiled-coil proteins (Supplementary Fig. 8b). Helix α3 does not contribute to the dimeric interface, but instead contains 10 residues that form an intramolecular interface with 14 residues of helices α2 and α4 (Supplementary Fig. 8c). To further investigate if the α-helical assembly of AlbB may be found in other characterized proteins, we performed a DALI[76] search using a monomer of AlbB as a query. Aside from mostly fragments of α-helical bundles found in unrelated structures, the search provided no structural alignments that were significantly similar to AlbB, suggesting that AlbB represents a distinct type of dimeric α-helical assembly.

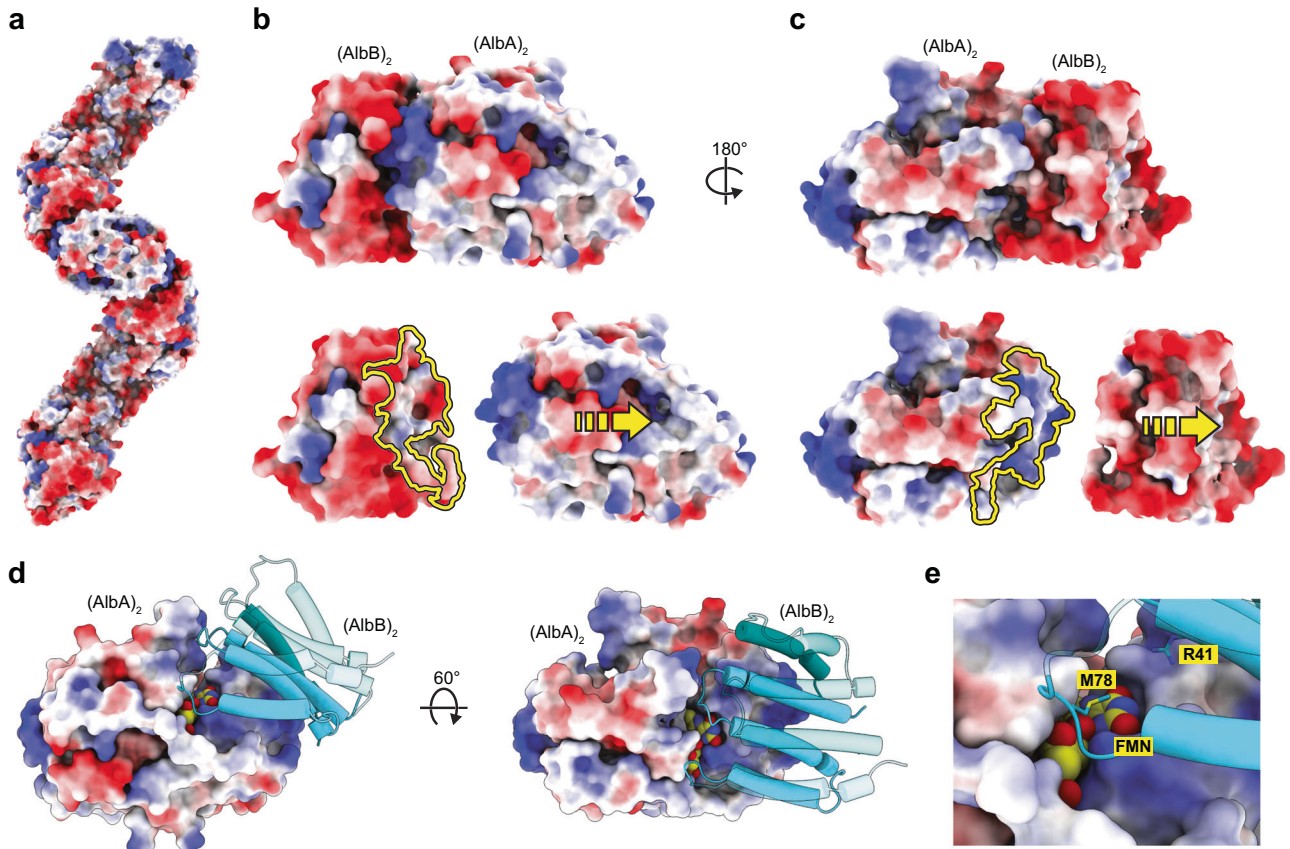

**Fig. 3 | Electrostatic interactions involved in filament assembly. a** Surface representation of the AlbAB filament with electrostatic coulombic potential displayed. **b** An AlbB$_2$-AlbA$_2$ tetramer is shown (top). Dimers pulled apart to highlight interaction surface (bottom). AlbB displays negative surface charges at the dimer-dimer interface (yellow outline). **c** An AlbA$_2$-AlbB$_2$ tetramer is shown (top). Dimers pulled apart to highlight interaction surface (bottom). AlbA displays positive surface charges at the dimer-dimer interface (yellow outline). **d** Dimer-dimer interface highlighting significant positive charge surrounding the FMN binding site. **e** Residues R41 and M78 of AlbB are located at the dimer-dimer interface forming parts of the FMN binding site.

## Interactions mediating AlbAB filament formation

The filament surface of AlbAB is characterized by periodic alternating charges, with AlbA displaying mostly neutral or positively charged surface areas, and AlbB possessing a mostly negatively charged surface (Fig. 3a–c). Residues at the AlbA$_2$-AlbB$_2$ dimer interface continue this pattern, with AlbA contributing mostly neutral or positively charged interface residues (Fig. 3c, d), and AlbB displaying mostly negatively charged interface residues (Fig. 3b). The most significant inter-dimer interactions occur between helix α3 of AlbA and helices α1 and α2 of AlbB (Fig. 3d). FMN cofactors are located at the AlbAB interface. They are primarily bound by AlbA, however, a number of AlbB residues do also directly interact with FMN (Fig. 3e).

The interface residues surrounding FMN are highly conserved throughout CDOAs and CDOBs (Fig. 4a, b and Supplementary Fig. 9a, b). In total, 20 residues of AlbA and 19 residues of AlbB contribute significantly to the interface (Fig. 4c, and Supplementary Fig. 9a, b). Within AlbA, residues R28, S54, N55, S139 are among the most conserved at the interface, likely due to their proximity to the FMN cofactor. AlbB residues E33, P34, Y37, and R41 are also located at the interface surrounding FMN and are almost completely conserved among all CDOBs. Notably, AlbA residue S54 and AlbB residue Y37 are both highly conserved and form a hydrogen bond above the *re* face of the FMN isoalloxazine ring.

## FMN binding and active site organization

As noted above, the FMN binding site is located at the interface of the AlbA$_2$ and AlbB$_2$ dimers (Fig. 5a, b), with clearly visible cryo-EM density corresponding to an 8α-*S*-cysteinyl linkage between the side chain thiol of C115 of AlbA and the C8 carbon of FMN. AlbA represents the only reported instance of an NTR-like protein with a covalently rather than a non-covalently bound FMN cofactor, which may function to increase the midpoint reduction potential of FMN[77]. Sequence alignments with other NTRs and CDOA proteins highlight that only CDOAs found in CDPS-encoding gene clusters contain a conserved C115 residue (Supplementary Fig. 9a–c). We identified four other CDOAs encoded in *Thermoprotea* archaea that contain a histidine residue in place of the C115 found in AlbA, potentially suggesting a covalent 8α-*N*$^1$-histidyl or 8α-*N*$^3$-histidyl linkage. All other identified CDOA proteins lack appropriately positioned residues known to covalently bind FMN. The FMN cofactor is further stabilized by extensive interactions with both AlbA and AlbB. The positively charged AlbA residues R24, R28, and R175 are highly conserved and coordinate the phosphate group of the FMN ribityl chain (Fig. 5c and Supplementary Fig. 9b, c). R28 forms an additional hydrogen bond with the N1-C2 = O2 position of FMN (Fig. 5c and Supplementary Fig. 9), an interaction which has been shown to increase the redox potential of FMN cofactors by stabilizing its reduced form (FMNH$_2$)[78]. Further, the backbone amide nitrogen of the AlbA residue G138 forms a hydrogen bond with the N5 atom of FMN (Supplementary Fig. 9), a conserved and likely necessary interaction for FMN catalysis previously observed in FMN-dependent dehydrogenases[78]. The nearby AlbA residues C135, P136, and V137 interact with the *si* face of FMN and are conserved in CDOA systems (Supplementary Fig. 9b, c). Residue M78 of AlbB further contributes to stabilizing the FMN cofactor.

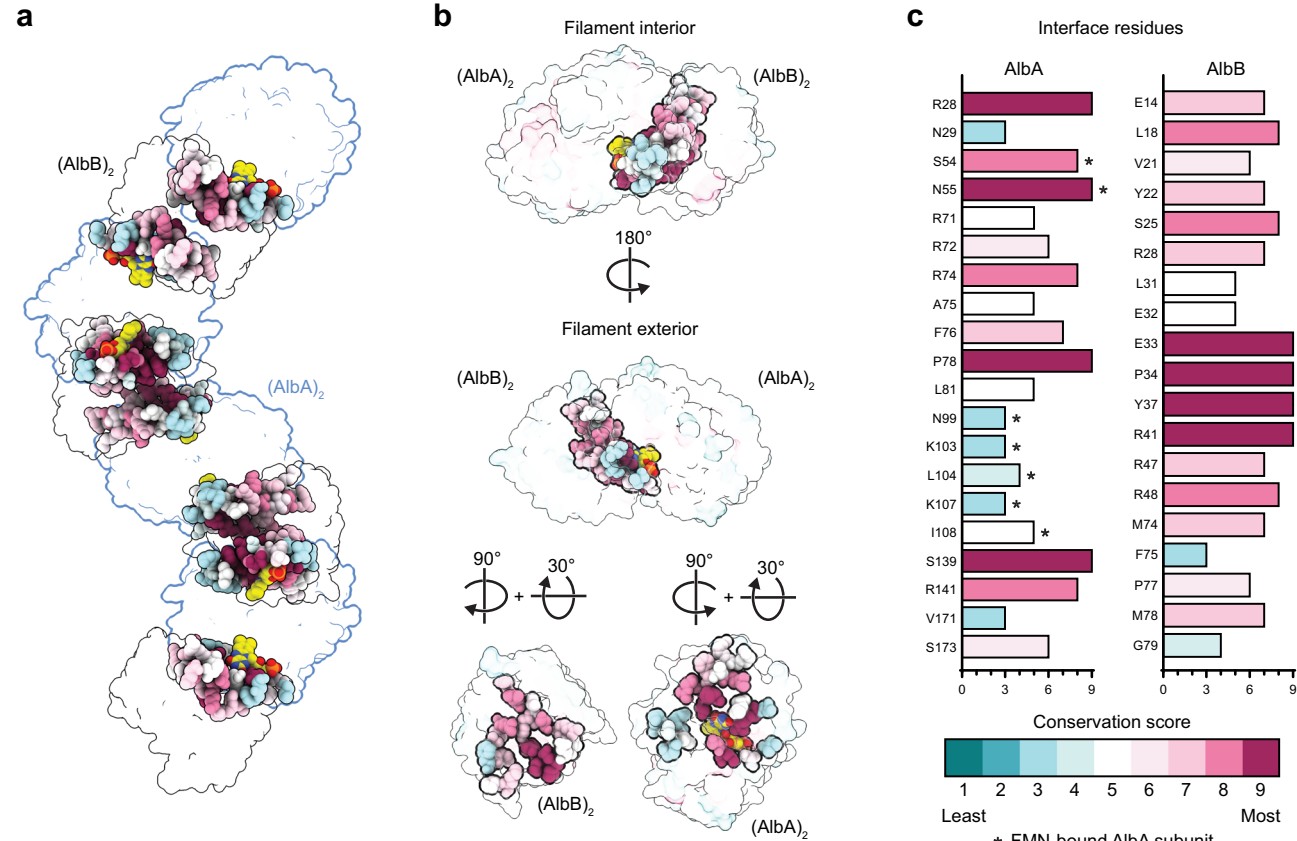

**Fig. 4 | Conservation analysis of the dimer-dimer interface in the AlbAB filament. a** AlbAB filament in transparent surface representation. The dimer-dimer interfaces are shown as spheres colored according to sequence conservation. FMN is shown as well colored according to atom type. AlbA$_2$ dimers are outlined in blue and AlbB$_2$ dimers in black. **b** Different perspectives of an AlbA$_2$-AlbB$_2$ tetramer highlighting amino acid conservation at the dimer-dimer interface and surrounding the FMN cofactor. **c** Conservation scores calculated with ConSurf[99–101] for the residues identified at the dimer-dimer interface. Only one AlbB subunit contributes to the interface while both AlbA subunits are involved. The residues of the AlbA subunit that carries the FMN cofactor at any given interface are highlighted (*).

The active site surrounding the FMN cofactor is exposed to solvent through three channels (Fig. 5d, e). One channel is accessible from the surface facing the interior long axis of the filament and is filled by the ribityl tail of FMN. Another channel is accessible at the interface of the AlbA$_2$ and AlbB$_2$ dimers and exposes the *re* face of the isoalloxazine ring. A third smaller channel is accessible near the dimer-dimer interface facing the exterior of the filament and exposes the N5 of FMN to solvent. Without substrate bound, the active site above the *re* face of FMN has an approximate volume of 1340 Å³ (Fig. 5e, f and Supplementary Fig. 10). The active site is capped by an interaction between the E2 extension of AlbA from residues T99 to I108 and residues 31-34 of AlbB. Residues within the E2 extension that are oriented toward the active site are not strongly conserved (Supplementary Fig. 5d). On the other hand, AlbB residues E33, P34, Y37, R41, M78, and G79 are strongly conserved and located near the active site, suggesting they may play a role in catalysis or in fine-tuning substrate positioning for catalysis.

The residues S54 of AlbA and Y37 of AlbB are located in hydrogen bonding distance of one another, directly above the FMN cofactor (*re* face) and are both highly conserved (Supplementary Fig. 10b). While the precise catalytic mechanism of AlbAB is currently unknown, an α-β bond in a cyclic dipeptide substrate must be oriented above the N5 nitrogen of FMN to allow for hydride transfer. This arrangement places Y37 or S54 in a likely orientation to function as a general base to carry out proton abstraction at the α or β position of the substrate, initiating α,β-dehydrogenation and hydride transfer to N5 of the FMN cofactor. Molecular docking of cyclo(ʟ-Phe-ʟ-Leu) into the active site of AlbAB

further supports this proposed mechanism, placing the substrate in orientations where the α or β carbons of cyclo(ʟ-Phe-ʟ-Leu) are located within 3 Å of the hydroxyl group of Y37 and within 4 Å of the N5 nitrogen of FMN (Supplementary Fig. 11a). Substrate orientations also suggest that the preference of AlbAB for hydrophobic residue-containing cyclic dipeptides may be based on the presence of an active site hydrophobic pocket formed by AlbA residues A58, L104, and I108, and the AlbB residue P34 (Supplementary Fig. 11b). However, more detailed structural studies will be needed to elucidate the details of AlbAB substrate preference.

To further elucidate AlbAB catalysis, we created the active site mutants S54A (AlbA) and Y37F (AlbB) to test our hypothesized mechanism. Both mutations resulted in significantly reduced catalytic activities, with S54A resulting in an approximately 62-fold decrease in catalytic activity, and Y37F exhibiting no measurable activity (Supplementary Fig. 11c). These results suggest a mechanism where S54 may interact with the Y37 side chain hydroxyl, allowing Y37 to function as the catalytic base responsible for proton abstraction from the α or β carbon of the cyclodipeptide substrate (Supplementary Fig. 11d). This proposed mechanism resembles the mechanisms suggested for some characterized NTR-like dehydrogenases like the thiazole/oxazole-modified microcin (TOMM) dehydrogenases that catalyze the formation of thiazoles and oxazoles during the biosynthesis of microcins and other peptide natural products[79–81]. A strongly conserved Lys-Tyr motif found at the tip of the central TOMM helix, was found to be important for catalysis, potentially acting as a general base to initiate dehydrogenation[79–81]. NTR-like dehydrogenase domains are also

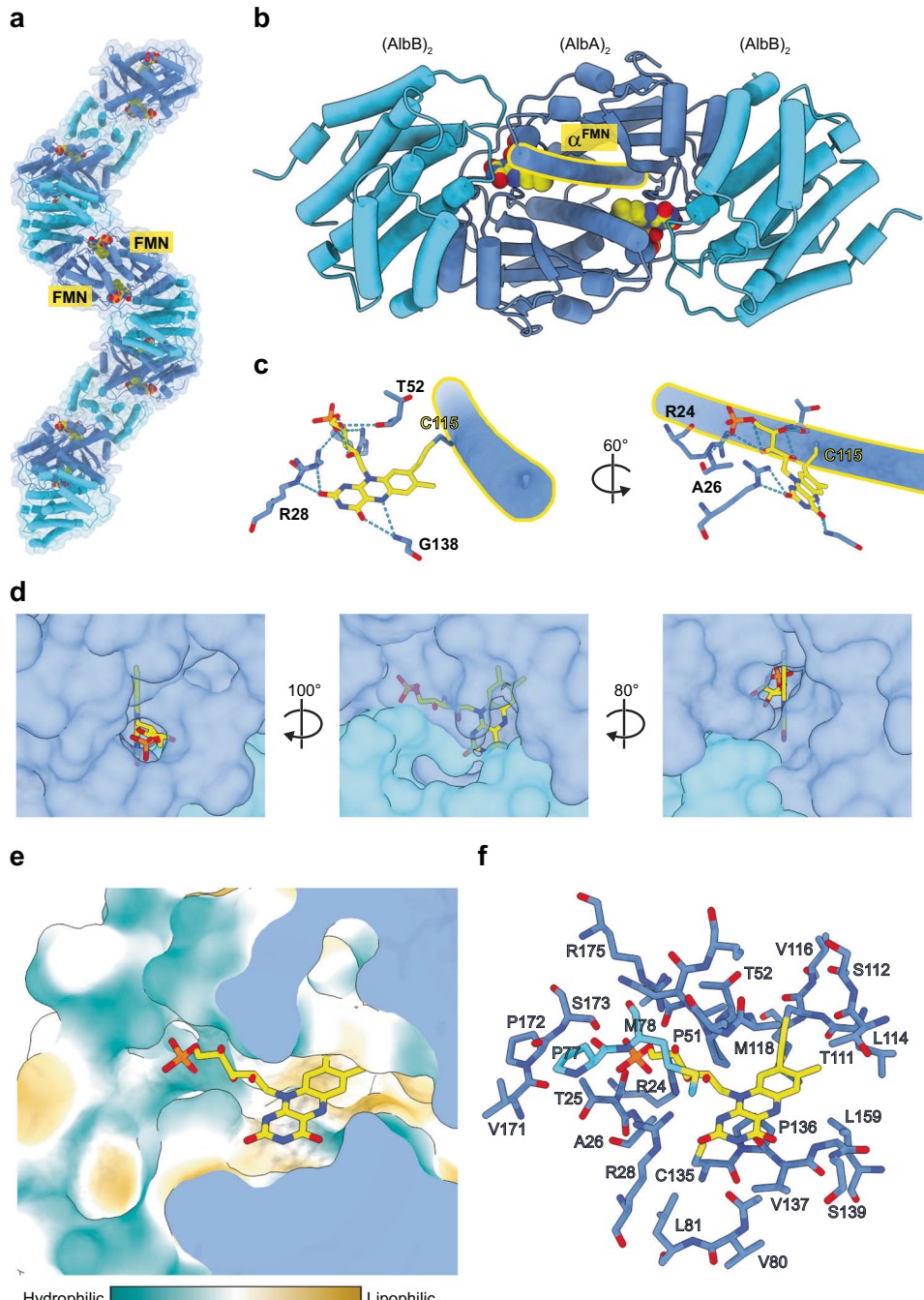

**Fig. 5 | Analysis of FMN binding and active site organization. a** AlbAB filament in ribbon highlighting the FMN cofactors (yellow) located at AlbA₂ (dark blue)-AlbB₂ (light blue) interfaces. **b** Ribbon representation showing an AlbA dimer interacting with two dimers of AlbB. The two interface FMN cofactors are shown (yellow). One AlbA α-helix containing C115 (α$^{FMN}$) – the site of covalent FMN attachment – is outlined (yellow) **c** FMN binding site showing covalent attachment to C115 via an 8α-*S*-cysteinyl linkage. The ribityl tail and isoalloxazine ring of FMN are further coordinated by multiple surrounding residues. Source data are provided as a Source Data file. **d** The three openings or channels to the FMN-containing active site are shown. **e** Surface representation colored by hydrophobicity of the FMN-containing active site. **f** Residues from both chains of AlbA and a single chain of AlbB contribute towards forming the FMN-binding pocket and active site located at each dimer-dimer interface.

present in multi-domain non-ribosomal peptide synthetases (NRPSs), functioning as oxidation domains for the formation of thiazolines and thiazoles during the biosynthesis of bleomycin[82,83], epothilone[83,84], colibactin[85], and indigioidine[86], likely by utilizing a conserved catalytic tyrosine to facilitate catalysis. Further, the NRPS-associated NTR-like tailoring enzyme BmdC involved in thiazole formation during bacilla-mide synthesis also possesses a conserved active site tyrosine thought to function as a general base during catalysis[87]. Our results suggest that AlbAB, similar to other characterized NTR-like dehydrogenases, utilizes a strongly conserved catalytic tyrosine during the oxidation of the α-β bond in cyclic dipeptide substrates.

## AlbAB filaments do not interact with AlbC or tRNA

Enzyme filaments can function as scaffolds to recruit other binding partners for various purposes including physical sequestration, modulation of activity, or improved multistep catalysis[88]. To investigate if

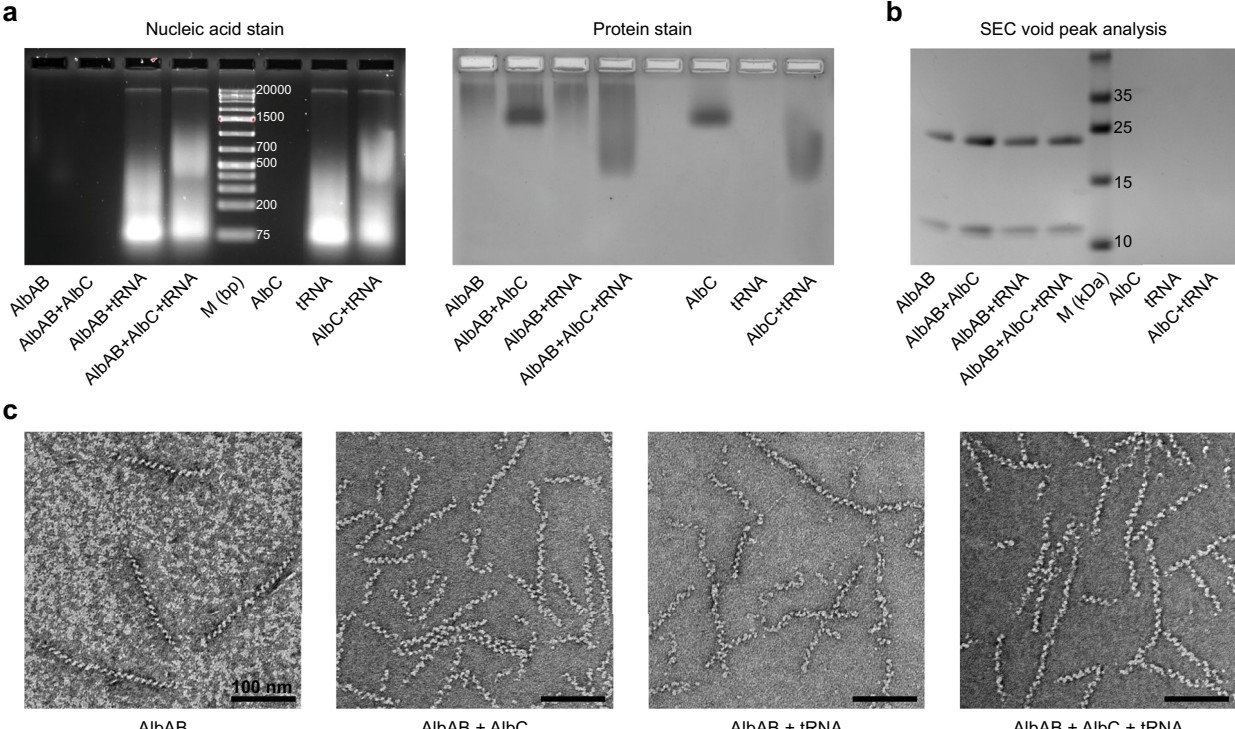

**Fig. 6 | Investigating the interactions of AlbAB with AlbC and tRNA. a** Agarose gel shift assays of AlbAB, AlbC, and *E. coli* total tRNA, as well as combinations thereof. Agarose gel stained for nucleic acids (GelCode Red) (left). The same agarose gel stained for protein (GelCode Blue). A gel demonstrating the purity of AlbC as well as complete agarose shift gels are provided in Supplementary Fig. 12a-c. Source data are provided as a Source Data file. This experiment was repeated independently five times. M: molecular weight marker. **b** SDS-PAGE gel of concentrated size-exclusion chromatography void fractions of AlbAB, AlbC, and

*E. coli* total tRNA, as well as combinations thereof. A Superdex 200 Increase column was used. AlbAB filaments exclusively elute in the void while neither AlbC nor *E. coli* total tRNA does. The complete SDS-PAGE gel is provided in Supplementary Fig. 12d. This experiment was repeated independently three times. **c** Negative stain TEM micrographs of the AlbAB-containing samples from part **b**. No samples demonstrate clear accessory binding of AlbC or tRNAs along AlbAB filaments. This experiment was repeated independently three times.

AlbAB filaments might interact with AlbC or tRNA – both necessary for the first step of albonoursin biosynthesis that generates cyclo(L-Phe-L-Leu) – we carried out agarose gel shift experiments. Such a direct interaction could function to facilitate the transfer of cyclo(L-Phe-L-Leu) from the AlbC active site directly to the active site of AlbAB. Gel shift assays did not show any interaction between AlbAB filaments and AlbC and/or total *E. coli* tRNA while showing a clear interaction between AlbC and tRNA (Fig. 6a). Interactions between CDPSs like AlbC and tRNA are expected and have been reported previously[34]. Gel filtration experiments with combinations of AlbAB, AlbC, and tRNA further showed no interaction between AlbAB and AlbC and/or tRNA (Fig. 6b). Negative stain TEM analysis of AlbAB-containing fractions from gel filtration runs did not show any apparent difference compared to freshly purified AlbAB filaments, and no protein or tRNA directly interacting with filaments could be observed (Fig. 6c).

## Discussion

Our AlbAB filament structure establishes the CDO family as enzyme filaments where filament formation is absolutely necessary for enzyme activity. While non-cytoskeletal filament-forming enzymes have been known for over 50 years, only recently have the molecular and physiological impacts of filament formation begun to be unraveled[88]. Filament-forming enzymes have been observed in bacteria and eukaryotes, including humans, and participate in diverse catabolic and anabolic pathways[88–90]. Filament formation can provide an additional layer of regulation, allowing the fine-tuning or spatial compartmentalization of enzyme activities within the cytosol[88,90]. Filament-forming enzymes generally possess a base level of catalytic activity in their non-filamentous forms, with filament formation resulting in increased or

decreased activity due to induced conformational changes[91]. AlbAB expands the functional diversity of enzyme filaments to include secondary metabolism and natural product biosynthesis, highlighting the ubiquitous use of filament formation as a strategy to spatially and functionally organize enzymes inside cells.

Based on our conservation analysis of residues found at the AlbA2-AlbB2 interface (Fig. 4), we hypothesize that filament formation is likely a requirement for activity – and potentially even solubility – in other CDOs as well. The requirement for co-expression is further supported by the fact that all CDOA and CDOB proteins are encoded by often overlapping genes with strong gene synteny. High sequence conservation in the FMN-containing active site, which is formed by residues of both AlbA and AlbB and located at the dimer-dimer interface, further supports filament formation as a general requirement for CDO activity. The CDOA/B fusion proteins identified in our phylogenetic analysis (Fig. 1 and Supplementary Fig. 1) may represent an evolutionary solution to the need for CDOA and CDOB co-expression and -assembly by ensuring correct subunit stoichiometry and more efficient formation of active CDO complexes.

The substrates of CDPS-associated CDOs are most likely cyclic dipeptides and their derivatives containing a 2,5-DKP scaffold. A number of CDOs found in CDPS gene clusters have been characterized to date and they often display some degree of substrate promiscuity[46,50,92]. Usually, CDOs display a preference for cyclic dipeptides containing aromatic or aliphatic amino acids over those containing charged or hydrophilic ones. This is in good agreement with the presence of multiple hydrophobic pockets and surfaces within the AlbAB active site (Fig. 5). In NTR-like proteins, substrate specificity is often determined by extensions to the minimal NTR

fold[72]. As CDOs do not exhibit any major extensions and are part of the minimal "hub" family of NTR-like proteins, binding specificity is likely determined by residues lining the active site at the dimer-dimer interface. Sequence conservation in the active site pocket declines with distance from the FMN cofactor (Supplementary Fig. 10c, d). High conservation of residues surrounding the FMN cofactor but not directly involved in FMN binding, may be responsible for the observed high specificity of CDPS-associated CDOs for the 2,5-DKP scaffold[47]. Decreased conservation of the mostly hydrophobic residues at the edges of the active site pocket, coupled with relatively large active site dimensions, may explain the ability of CDOs to accept a broad range of different cyclic dipeptide substrates.

We discovered that the majority of CDO-like proteins are found in non-CDPS-associated gene clusters (Fig. 1 and Supplementary Fig. 1). Given the prevalence of predicted hydantoin- and allantoin-modifying enzymes in these gene clusters, these CDO-like enzymes may utilize substrates structurally related to allantoins or hydantoins which contain a 2,4-imidazolidinedione scaffold, structurally reminiscent of the 2,5-DKP backbone found in cyclic dipeptides. As hydantoins and allantoins are known intermediates in purine metabolism, non-CDPS-associated CDO-like enzymes could participate in nucleobase-related metabolic pathways[52,53]. Another possibility is that newly identified CDO-like enzymes are involved in the biosynthesis of so far unknown secondary metabolites containing hydantoin- or allantoin-related functional groups[54]. We cannot exclude the possibility that non-CDPS-associated CDO-like enzymes may not function as oxidases or dehydrogenases. The broad substrate repertoire and various reactions performed by NTR-like enzymes makes it difficult to predict the biological functions of the non-CDPS-associated CDO-like enzymes. Further experimental investigation is necessary to elucidate the function of these enzymes.

The results presented here provide molecular-level details on CDO structure and expand the likely functions of CDO-like enzymes beyond CDPS-dependent cyclic dipeptide biosynthesis. Our AlbAB structure represents another addition to the relatively limited number of filamentous enzymes and further underscores the prevalence of filament formation as an organizational strategy. Our structure will facilitate future mechanistic investigations into CDO function and provide useful molecular-level detail for utilizing CDOs as enzyme catalysts in biocatalysis and chemoenzymatic synthesis.

## Methods
### Bioinformatic and computational analysis
Sequences for CDOA and CDOB were obtained using the Enzyme Function Initiative Enzyme Similarity tool (EFI-EST)[93,94] to generate an initial sequence similarity network (SSN) of only non-fragmented CDOB proteins from the UniProtKB[95] database in the Pfam[55] family PF19585. This search retrieved 304 accessions corresponding to CDOB proteins. These accessions were then used as an input for the Enzyme Function Initiative Genome Neighborhood Tool[93,94], which was used to retrieve the accessions for CDOA, CDPS, and other proteins encoded in CDO operons. Genomes that did not contain both a complete CDOA and CDOB sequence were removed from the dataset, resulting in a final dataset containing 274 operons (Supplementary Data 1).

EFI-EST was used to generate an SSN of CDOAB systems by concatenating the CDOA and CDOB proteins found in each operon. CDOAB fusion proteins were not concatenated and were included in the dataset without any modification. Cytoscape v3.10.0[96] was used to remove edges with less than 46% sequence identity, and the SSN was visualized and annotated using the yFiles Organic Layout application in Cytoscape (Supplementary Data 2).

To generate a phylogenetic tree of all identified CDOs, the concatenated sequences from the SSN were first aligned using MAFFT v7[97] (MAFFT.cbrc.jp) with default parameters (Supplementary Data 3). The sequence alignment was then assembled into a phylogenetic tree by

forwarding to the Phylogeny tool on the MAFFT online server with the following parameters: 274 sequences, 667 total sites, 97 gap-free sites, 34 conserved sites, NJ model (all of gap-free sites), JTT substitution model, ignore heterogeneity among sites ($\alpha=\infty$), bootstrap resampling = 1000. The phylogenetic tree was then visualized as a radial tree, ignoring branch lengths, and annotated using iTOL v6[98].

To generate the phylogenetic tree of all NTRs, MAFFT v7 was used to align AlbA and 54 other NTR protein sequences obtained from the PDB that encompassed various NTR families (Supplementary Data 4). The multiple sequence alignment (MSA) was then assembled into a phylogenetic tree by forwarding to the Phylogeny tool on the MAFFT online server with the following parameters: 55 sequences, 854 total sites, 97 gap-free sites, 5 conserved sites, NJ model (all of gap-free sites), JTT substitution model, ignore heterogeneity among sites ($\alpha=\infty$), bootstrap resampling = 1000. The phylogenetic tree was then visualized as a radial tree, ignoring branch lengths, and annotated using iTOL v6.

Conservation analysis was performed via ConSurf[99–101] using the AlbAB model as an input against MSAs of only CDPS-associated CDOs, all identified CDOs, and a curated list of NTR with available structural information (Supplementary Data 5 and 6).

### Molecular cloning
Genes encoding AlbAB and AlbC were ordered as gBlocks from Integrated DNA Technologies (IDT) (Supplementary Table 2). A *Streptomyces coelicolor* overexpression plasmid containing AlbAB was constructed via Gibson assembly by mixing the AlbAB-encoding gBlock and a pGM1190 vector[102] digested with NdeI and BamHI restriction enzymes with Gibson assembly master mix (NEB) for 20 min at 50 °C. AlbA S54A and AlbB Y37F mutants were prepared by similar Gibson assembly using complementary primer sets that contained the respective mutations. N-terminally His-tagged AlbB and AlbC plasmids, containing tobacco etch virus (TEV) cleavage sites to remove the tag, were constructed by PCR amplifying the respective genes from the AlbAB-encoding pGM1190 vector for AlbB, or from a gBlock obtained from IDT for AlbC, followed by Gibson assembly with a pETDuet-1 vector utilizing multiple cloning site 2. Assembled plasmids were then transferred into electrocompetent *E. coli* BL21 (DE3) cells via electroporation. Plasmids were confirmed by Sangar sequencing (Eurofins). All PCR steps were carried out using Q5 DNA polymerase (NEB).

### Protein expression
Plasmid conjugation and protein expression in *S. coelicolor* (ATCC #BAA-471, also known as *Streptomyces violaceoruber* strain John Innes Centre M145) was performed as described previously[103]. Briefly, the modified pGM1190 vectors encoding AlbAB, AlbA-S54/AlbB, or AlbA/AlbB-Y37F were introduced into *E. coli* ET12567/pUZ8002 cells via transformation by heat shock and maintained on LB agar plates containing 50 μg/mL apramycin, 30 μg/mL chloramphenicol, and 50 μg/mL kanamycin. Plasmid-containing colonies were used to inoculate 20 mL cultures of LB medium containing the same antibiotics, which were grown overnight at 37 °C with shaking at 200 rpm. Cells were harvested by centrifugation at 3800 x g, washed twice in sterile LB to remove antibiotics, and suspended in 2 mL of LB. 200 μL of the plasmid-containing *E. coli* ET12567/pUZ8002 cells were mixed with 500 μL of freshly prepared *S. coelicolor* spore stock contained in GYM media, plated on MS-agar containing 10 mM MgCl$_2$, and incubated overnight at 30 °C. After 18 h of incubation, plates were overlayed with 1 mg apramycin and 1 mg of nalidixic acid and incubated at 30 °C for 3–5 days until spores were visible. To further select for plasmid-containing *S. coelicolor* colonies, spores were picked and struck out on an additional plate of MS-agar containing 50 μg/mL of apramycin and incubated a further 3–5 days until colonies became visible. A single positive colony from this plate was then picked and spread on an additional MS-agar plate, incubated at 30 °C for 3–5 days.

The spores were scraped from the plate and used to inoculate cultures of 500 mL of YEME media containing 50 µg/mL of apramycin in a 2 L baffled flask containing glass beads for enhanced aeration and prevention of mycelium formation, which were incubated at 30 °C with shaking at 200 rpm for 24–36 h until $OD_{600}$ reached between 0.4 and 0.8. 20 µg/mL of thiostrepton was added upon reaching the desired optical density, and cultures were further shaken for 72 hours at 30 °C to allow protein overexpression. Cells were then harvested via centrifugation at 8000 x g and stored at −20 °C.

His-tagged AlbB and AlbC encoding plasmids were inserted into *E. coli* BL21 cells by transformation via electroporation and plated on LB agar plates containing 100 µg/mL of ampicillin. A single colony of each construct was used to inoculate respective 10 mL cultures of LB containing 100 µg/mL ampicillin and the cultures were grown overnight with shaking at 200 rpm at 37 °C. These cultures were then used to inoculate 500 mL cultures of LB containing 50 µg/mL ampicillin, which were grown in 2 L baffled flasks at 37 °C with shaking until they reached an optical density between 0.5 and 0.8, and subsequently induced with addition of 0.5 mM IPTG. Proteins were expressed with shaking at 18 °C for 18 hours, harvested via centrifugation at 8000 x g, and the cell pellets stored at −20 °C.

## Protein purification

AlbAB filaments were purified by resuspending 1.5 g of a *S. coelicolor* frozen cell pellet in 7.5 mL of Buffer A, containing 150 mM NaCl, 20 mM Tris at pH 7.5. 1 mg/mL lysozyme, 250 units of Benzonase Nuclease (Millipore Sigma), and a protease inhibitor cocktail with a 1x concentration of 10 µM leupeptin, 14 µM E-64, 0.4 µM aprotinin, 0.4 mM AEBSF, 114 µM bestatin were added to Buffer A prior to lysis. The resuspended cells were lysed by sonication at 96 W for a total time of 3 min on ice using a Model 120 Sonic Dismembrator from Fisher Scientific, Inc. (USA). The lysate was then clarified by centrifugation at 21,000 x g for 10 min. Ammonium sulfate was added to the clarified supernatant to 30% saturation and incubated with gentle rocking at 4 °C for approximately 45 min, after which the sample was clarified by centrifugation at 10,000 x g for 10 min. The AlbAB-containing pellet was resuspended in 2 mL of Buffer A and heated to 50 °C for 20 min, followed by centrifugation at 10,000 x g for 10 min, which resulted in a yellow-orange pellet containing AlbAB along with denatured proteins. The pellet was washed three times by resuspending the pellet in 500 µL of Buffer A followed by centrifugation at 5000 x g for 10 min. The second and third washes were pooled together, concentrated to 0.5 mL in an Amicon Ultra-15 centrifugal filter with a 100 kDa cutoff, and run over a Superose 6 Increase 10/300 GL column using Buffer A (150 mM NaCl, 20 mM Tris pH 7.5) as the running buffer for further purification. AlbAB-containing fractions were pooled, concentrated to 2.4 mg/mL, frozen in liquid nitrogen, and stored at −80 °C. The same method was used to purify the AlbA S54A/ AlbB mutant. Prior to Superose 6 Increase 10/300 GL, the AlbA/AlbB Y37F mutant was further purified by ion exchange using a HiPrep DEAE FF 16/10 column to remove contaminating prodigiosin dye produced by *S. coelicolor* during overexpression, otherwise the purification protocol was the same as the other AlbAB proteins.

His-tagged AlbB and AlbC proteins were purified by lysing cell pellets in Buffer B containing 300 mM sodium chloride, 20 mM Tris pH 7.5, and 20 mM imidazole. 10 µM leupeptin, 14 µM E-64, 0.4 µM aprotinin, 0.4 mM AEBSF, 114 µM bestatin, and 0.5 mg/mL lysozyme were added prior to lysis. The cells were sonicated at 96 W for a total time of 3 min on ice. The lysates were clarified by centrifugation at 21,000 x g for 10 minutes and immediately loaded onto a His-Trap FF affinity purification column. The column was washed with 10 column volumes of buffer B, followed by 5 column volumes of buffer B containing 40 mM imidazole. Proteins were eluted with 5 column volumes of Buffer B containing 250 mM imidazole. 2 mg of TEV protease and 0.3 mM TCEP were added to the purified protein samples, and the

proteins were dialyzed against 1 L of 100 mM NaCl, 20 mM Tris pH 7.5, 0.3 mM TCEP at 4 °C for 18 h with spinning. Following dialysis, the protein samples were spun for 10 min at 10,000 x g to remove any denatured protein. The clarified protein samples were then flowed over another His-Trap FF affinity purification column to remove any remaining His-tagged protein. The columns were washed 5 column volumes of Buffer B containing 40 mM imidazole. The flowthrough and wash samples were pooled, concentrated and desalted into 150 mM sodium chloride, 20 mM Tris pH 7.5 using Amicon Ultra-15 centrifugal filters with a 3 kDa MW cutoff for AlbB, or a 10 kDa MW cutoff for AlbC. The concentrated proteins were then immediately drop frozen in liquid nitrogen and stored at −80 °C.

## Sample preparation and cryo-EM data collection

Grids of AlbAB were prepared by applying 3.5 µL of freshly prepared protein at concentration of 0.75 mg/mL to glow-discharged Quantifoil R2/1, 200 mesh copper holey carbon grids (EMS, Cat# Q225CR1). The grids were then frozen by plunging into liquid ethane using an FEI Vitrobot Mark IV with the following parameters: temperature 22 °C, humidity 100%, blot force 5, blot time 2 seconds. The grids were clipped and stored in liquid nitrogen until data collection.

Data was collected using a ThermoFisher Krios G4I cryo-transmission electron microscope operating at 300 kV and equipped with a Gatan K3 Direct Detector with BioQuantum imaging filter housed at the University of Michigan Life Sciences Institute. SerialEM was used to select targets and acquire movies with the following settings: defocus range −1 to −2.5 µm, dose 50.1 e⁻/Å², magnification 105,000x, exposure time 2.03 s with 40.6 ms per frame. 3015 movies were collected in total from a single grid.

## Cyro-EM data processing and model building

All data processing was carried out using CryoSPARC v4.2.1[104,105]. 3,015 movies were motion corrected using Patch Motion Correction, followed by contrast transfer function (CTF) estimation using Patch CTF Estimation. Movies with CTF fits worse than 5 Å were removed from the dataset, resulting in 2842 movies. 834 particles were picked manually and used to generate initial templates via 2D classification. Particles were then picked from all movies by using Template Picker with a particle size of 100 Å, resulting in 1,202,923 particles which were extracted with a box size of 360 pixels. The selected particles were then sorted using two rounds of 2D classification with 100 classes. Good classes were selected, resulting in 795,765 particles, which were then used as input for a Helix Refine job with estimated helical twist of 118° and estimated helical rise of 44 Å, resulting in a map with a global resolution of 3.14 Å and an optimized helical twist of 119.969° and optimized helical rise of 46.063 Å. The map was then used as an input for a C1 Local Refinement job using a mask that included a dimer of AlbA and two surrounding dimers of AlbB, which yielded an improved map with a global resolution of 2.78 Å.

An initial model containing two dimers of AlbB and one dimer of AlbA was generated using AlphaFold2 with MMseqs2 on ColabFold v1.5.2[106]. The starting model was docked into the map generated by Local Refinement and was refined iteratively by alternate rounds of real-space refinement using Phenix v1.20.1-4487[107,108] and manual refinement in Coot v0.9.8.1[109,110]. During refinement, the model was further reduced to include only a dimer of AlbA and a dimer of AlbB to better represent the biologically-relevant asymmetric unit of the filament.

## Filament stability assays

To investigate the stability of AlbAB filaments under various pH and salt conditions, purified AlbAB filaments were diluted to 0.2 mg/mL in Buffer A, spun for 10 min at 10,000 x g and 4 °C. Then, 40 µL of AlbAB sample was dialyzed for 18 h against 50 mL of either of the following buffers: 0 mM NaCl, 20 mM Tris pH 7.5; 25 mM NaCl, 20 mM Tris

pH 7.5; 150 mM NaCl, 20 mM Tris pH 7.5; 1 M NaCl, 20 mM Tris pH 7.5; 150 mM NaCl, 20 mM sodium citrate pH 4.1; 150 mM NaCl, Bis-Tris pH 5.5; or 150 mM NaCl, 20 mM Bis-Tris pH 8.5. Following dialysis, the protein samples were diluted to 0.05 mg/mL in their respective dialysis buffers and 3.5 µL of the protein samples were applied to freshly glow-discharged formvar-reinforced carbon grids (EMS: FCF-200-AU-EC), washed with water, and stained by application of 5 µL of 0.2% uranyl formate. Micrographs were collected using an FEI Morgagni transmission electron microscope operating at 100 kV equipped with a Gatan Orius SC200 CCD detector housed at the University of Michigan Life Sciences Institute.

### SDS-PAGE analysis of protein samples
Protein samples for SDS-PAGE analysis were prepared by mixing with Invitrogen NuPAGE LDS Sample Buffer (4x) (Invitrogen, Cat# NP0007) containing 350 mM 2-mercaptoethanol and heat-denaturing for 2 minutes at 95 °C. The protein samples were then run on NuPAGE 4–12% Bis-Tris gels (Invitrogen, Cat# NP0323BOX) using Novex NuPAGE MES SDS running buffer at 200 V for 35 min. Gels were stained using ReadyBlue protein gel stain (Sigma-Aldrich, Cat# RSB-1L) or Coomassie Brilliant Blue R-250 (Thermo Scientific, Ref # 20278) followed by de-staining with water or de-staining buffer. Gels were imaged using a Bio-Rad Chemidoc Imaging station.

### Dynamic light scattering (DLS) analysis
DLS measurements were carried out using an Unchained Labs (USA) Uncle instrument at 25 °C. Purified samples of AlbAB and AlbB were diluted to 0.1 mg/mL and 1 mg/mL, respectively in 150 mM NaCl, 20 mM Tris pH 7.5. Samples were centrifuged at 5000 x g for 10 min at 4 °C immediately prior to DLS measurements.

### Enzyme assays
Assays were performed in triplicate following Gondry et al.[111]. Briefly, 100 µL assay mixtures containing 0.25 µM purified AlbAB, 100 mM Tris pH 8.0, 1 mM 4-hydroxyphenylacetic acid (Sigma-Aldrich, Cat# H50004-5G), 0.1 U horseradish peroxidase (Sigma-Aldrich, Cat# P8250-5KU), and varying concentrations of cyclo(L-Phe-L-Leu) (Chem-Impex International, Inc. Cat# 11054) were prepared in black 96-well plates (Corning Ref: 3650). Enzymatic reactions were initiated by the addition of cyclo(L-Phe-L-Leu). Production of $H_2O_2$ was measured using a Synergy H1 plate reader by monitoring the oxidation of 4-hydroxyphenlyacetic acid into the fluorescent product 6,6'-dihydroxy-(1,1'-biphenyl)–3,3'-diacetic acid ($\lambda_{ex} = 318$ nm, $\lambda_{em} = 405$ nm), with an increase of fluorescence emission at 405 nm directly relating to the increase in $H_2O_2$ concentration. Reactions for wild-type AlbAB were monitored for 30 min at 30 °C with shaking between measurements that were taken every 12 s in sweep mode. The reactions for the AlbA-S54A/AlbB and AlbA/AlbB-Y37F mutants were monitored for 120 min using the same parameters. Fluorescence intensities were fit to a standard curve of known concentrations of $H_2O_2$ to quantify the extent of oxidation. GraphPad Prism v9 was used to calculate velocities by linear regression during the first 90 s of the reaction of the wild-type AlbAB. Because the rate of AlbA-S54A/AlbB was so much lower than wild-type, velocities were calculated from datapoints that were collected between 800 and 1340 s for reactions containing 10–100 µM of cyclo(L-Phe-L-Leu) and 1424–2000 s for 5 µM of cyclo(L-Phe-L-Leu). Datapoints for AlbA/AlbB-Y37F were collected between 400 and 1000 s. The velocities for AlbAB and AlbA-S54A/AlbB were then fit to a Michaelis-Menton non-linear regression model in GraphPad Prism v9 to derive the kinetic parameters.

### Molecular docking
Molecular docking was performed using the Webina server[112]. AlbAB was used as a receptor input and an atomic model of cyclo(L-Phe-L-Leu) obtained from as a.SDF file from the PubChem database (PubChem CID:

7076347) was used as a ligand input. The search grid center was placed at the following coordinates: X:162, Y:169, Z:144 with a box size of X:16, Y:13, Z:17. All other settings were left as default. Both ligand and receptor were appropriately protonated (pH 7.4) during the Webina workflow.

### AlbAB, AlbC, and tRNA interaction experiments
AlbAB interactions with AlbC and tRNA were first tested by adding mixed samples to NativePAGE loading dye (Invitrogen, Cat# BN20032) followed by loading onto a 1.5% agarose gel prepared with fresh 1x TAE buffer. The agarose gel was run for 50 min at 90 V and 4 °C. tRNA was visualized by staining the gel for 30 min at room temperature with a 1x mixture of GelCode Red (EMD Millipore, Cat# SCT123). Protein was subsequently visualized by staining with GelCode Blue stain (Thermo Scientific, Cat# 1860957) for 18 h and then de-staining with water for an additional 18 h.

To further test whether AlbAB interacts with AlbC or tRNA, mixed samples were prepared containing 10 µM of all components. Mixed samples were then subjected to size-exclusion chromatography using a Superdex S-200 column. AlbAB was found in the void fractions and as such, the void fractions were collected from each run and concentrated to 50 µL. 7.5 µL of each concentrated sample were then analyzed by SDS-PAGE using NuPAGE 4–12% Bis-Tris gels. AlbAB-containing samples were then diluted to 0.05 mg/mL and visualized by negative stain TEM.

### Reporting summary
Further information on research design is available in the Nature Portfolio Reporting Summary linked to this article.

## Data availability
The cryo-EM map and structural model of AlbAB have been deposited and are publicly available in the Electron Microscopy Data Bank under accession number EMDB-42114 and Protein Data Bank with ID 8UC3. Sequence similarity networks and sequence alignments are supplied as Supplementary Data. Source data are provided with this paper.

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

## Acknowledgements

We acknowledge funding from the NIH (R35GM133325). We acknowledge support by the UM Research Scouts program. Research reported in this work was supported by the University of Michigan Cryo-EM Facility (U-M Cryo-EM). U-M Cryo-EM is grateful for support from the U-M Life Sciences Institute and the U-M Biosciences Initiative. Molecular graphics and analyses were performed using UCSF ChimeraX developed by the Resource for Biocomputing, Visualization, and Informatics at the University of California, San Francisco, with support from the National Institutes of Health R01GM129325 and the Office of Cyber Infrastructure and Computational Biology, National Institute of Allergy and Infectious Diseases.

## Author contributions

M.P.A. and T.W.G. designed the project. M.P.A. and T.W.G. conducted the bioinformatic and phylogenetic analyses. M.P.A. conducted all laboratory experiments. M.P.A. collected cryo-EM data, and M.P.A. and T.W.G. processed and analyzed the cryo-EM data. M.P.A. built the structural model. M.P.A. and T.W.G. wrote the manuscript. T.W.G. oversaw the project in its entirety.

## Competing interests

The authors declare no competing interests.
