## [Peer Review File · Nature Communications]

Cyclodipeptide oxidase is an enzyme filamentREVIEWER COMMENTS

Reviewer #1 (Remarks to the Author):

In their manuscript titled Cyclodipeptide oxidase is an enzyme filament, Andreas and Giessen present structural and functional analyses of the cyclodipeptide oxidase (CDO) AlbAB. To date, few CDOs have been characterized in depth, and all employ CDOA (nitroreductase-like) and CDOB (no known homology) subunits to perform O₂-dependent α,β -dehydrogenations on cyclodipeptides (CDPs). Diversification of CDPs is attractive, given their biologically relevant activities and their incorporation into various, more chemically complex drug molecules.

Through bioinformatics analyses presented in the manuscript, the team have considerably expanded the number of known CDOs, identified previously undescribed CDOA/CDOB fusions, and found numerous CDOs in non-CDPS-encoding gene clusters. Interestingly, they also observed CDO sequences in anaerobic bacteria, despite the expectation of O₂ as the terminal electron acceptor. In a significant stride forward with these previously intractable enzymes, the authors next expressed and purified AlbAB from the heterologous host *Streptomyces coelicolor*. Consistent with its appearance in the size-exclusion chromatography void volume, the protein was observed in long filaments by negative stain TEM analysis; these filaments were found to be stable under a range of pH values and ionic strengths.

The team also performed the first in vitro activity and kinetic analyses on a purified CDO, testing AlbAB against its native substrate cyclo(L-Phe-L-Leu) through an established two-enzyme reaction with horseradish peroxidase. Notably, AlbA could not be individually isolated, and AlbB was inactive on its own and failed to form filaments. However, AlbAB filaments retained their filamentous structure under reaction conditions, as shown by TEM. Formation of these filaments was further supported by size-exclusion chromatography and dynamic light scattering experiments.

Next, the authors carried out structure determination of AlbAB using single-particle cryo-EM. Here, they observed that dimers of each of AlbA and AlbB form a heterotetrameric unit along the helical filament. Moreover, from the cryo-EM density, FMN could be identified as the flavin cofactor, covalently linked to each AlbA monomer. The structures of AlbA and AlbB, their homodimer/heterodimeric interfaces, and FMN binding sites were subsequently discussed in detail. Finally, through agarose gel shift assays and size-exclusion chromatography, the authors demonstrated that neither the CDPS AlbC nor tRNA form a stable complex with AlbAB filaments.

As a whole, I find the manuscript clearly and elegantly written, the analysis comprehensive, and the results beautifully illustrated. The discovery and characterization of filament-forming enzymes involved in secondary metabolite production is unexpected yet exciting in terms of biocatalytic potential,

especially as CDPs are useful chemical precursors with important bioactivities. I suggest the submission to be highly suitable for publication in Nature Communications once the following points are addressed.

1. Can the structural results of the authors be used to better understand the substrate specificity and/or catalytic mechanism, and thus the potential of AlbAB for biocatalysis? For example, in silico docking (either manual or automated) of enzyme and substrate or product could yield insight into this aspect. Indeed, the authors have stated that “an α - β bond in a cyclic dipeptide substrate must be oriented above the N5 nitrogen of FMN to allow for hydride transfer”; they also suggested a catalytic role for Tyr37 of AlbB, which is reasonably given a similar role in homologous enzymes. These constraints should (hopefully) allow the substrate to be reasonably placed within the active site, especially as the amide NH and C=O atoms are likely to form specific H-bonds to nearby atoms.

2. Preparing a few active site mutations and repeating activity assays would be useful in supporting the assignment of substrate-binding/catalytic residues. This could be particularly powerful when combined with docking results.

3. What does a DALI search (<http://ekhidna2.biocenter.helsinki.fi/dali/>) yield for AlbB? Are there closely related and/or interesting structures, despite the lack of sequence homology?

4. What is the rationale for testing activity at pH 4.1 (see line 155)?

5. What are the expected absorbance maxima for a non-covalently associated FMN molecule (see line 161)?

6. Other corrections:

a. Line 147: Please change “equimolar” to “approximately equimolar”, as concentration estimation by SDS-PAGE is not very accurate.

b. Line 365: “elude” should be “elute”.

c. Line 384: Something is missing here – possibly “and” in “...filaments, [and] no protein...”.

d. Line 566: “5,000 x G” should read “5,000 x g”.

e. Line 607: Please define the abbreviation “CTF”.

f. Lines 623-624: Please provide literature references for the Phenix and Coot software packages.

g. Figure S3 caption: Please define 4-HPA.

Reviewer #2 (Remarks to the Author):

In this work, Andreas and Giessen present cryo-EM and other data supporting the AlbAB cyclodipeptide oxidase (CDO) as a heterooligomeric enzyme filament. They also provide bioinformatics analyses of a collection of approximately 275 other CDOs, suggesting similarities in structure and function of these CDOs with AlbAB. The results of this work should be of interest to the biosynthesis community. Overall, their data are compelling and the evidence is well presented. There are a few concerns as noted below:

—The flow of Introduction may be improved by breaking the first paragraph into two paragraphs, perhaps with a break point after the CDPS/RCDPS discussion.

—Fig 1h lacks error bars and does not indicate replication of the kinetics assay. Adequate replicates should be incorporated.

—Under “AlbAB is a filament-forming CDO with covalently bound flavin cofactors” section, it is mentioned that purified AlbAB has a “very high apparent molecular weight.” This statement would be more impactful if a weight range were stated.

—Paragraph 4 of Discussion: Since the experimental data of the paper focus on AlbAB rather than CDO-like proteins in non-CDPS clusters, much of this paragraph is speculative in nature and overly redundant of bioinformatics discussions presented in the Results. This paragraph should be made more focused and succinct.

Reviewer #3 (Remarks to the Author):

The manuscript submitted by Andreas and Giessen focuses on the structural and functional characterization of AlbAB, a cyclodipeptide oxidase (CDO) from *Streptomyces noursei* involved in albonoursin biosynthesis. The authors report that AlbAB forms a megadalton heterooligomeric enzyme filament with covalently bound flavin mononucleotide cofactors, and they emphasize the importance of filament formation for enzyme activity. The study suggests that AlbA-AlbB interactions are conserved, indicating a potential generalization to all CDO-like enzymes functioning as enzyme filaments. The work addresses historical challenges in studying CDOs and provides insights into their structural and functional aspects. The authors anticipate implications for biocatalysis and chemoenzymatic synthesis. Overall, the manuscript contributes to the understanding of CDOs, marking a notable advancement in the field of enzymology and natural product biosynthesis.

In this work, single-particle cryo-electron microscopy (cryo-EM) played a pivotal role in unveiling the molecular structure and assembly of the AlbAB filament. The technique was specifically employed to provide detailed insights into the architecture of the cyclodipeptide oxidase (CDO) AlbAB, allowing the visualization of the arrangement of AlbA and AlbB subunits within the heterooligomeric enzyme filament. This detailed structural information was crucial for understanding the organization of the AlbAB filament, offering a foundation for future investigations into the functional mechanisms of CDOs and providing valuable insights into their catalytic activity and substrate interactions.

The authors have undertaken a helical reconstruction employing the single-particle analysis implementation within CryoSPARC, coupled with a local refinement approach. Their meticulous approach to the structural analysis reflects a technically rigorous examination. The resulting characterization demonstrates a thorough and detailed understanding of the structural intricacies, showcasing a commendable level of precision in their investigative methods.

The manuscript is well-crafted, demonstrating clarity in its writing style that enhances readability. Despite the intricacy of the analysis, the text remains reasonably approachable for individuals who may not be specialists in the field. Only a few minor changes are needed to further refine the document.

Minor Comments

Pp7 ll 185-186: "Initial 2D class averages displayed clear periodicity highlighting two distinct subunits along the length of the filament (Fig. 2a)"

The presence of the two distinct subunits is not immediately apparent in the initial 2D class averages. To enhance clarity, especially for non-expert readers, it is recommended to highlight these subunits in at least one of the 2D averages.

Pp7 LI 183-198. Each AlbA2 dimer and AlbB2 dimer appears to exhibit a local 2-fold symmetry within the subunits that constitute the dimer. If confirmed, it should be described in the text.

POINT-BY-POINT RESPONSE TO REVIEWERS:

Author replies shown in blue

Yellow background indicates new text added to the manuscript

Reviewer #1:

In their manuscript titled Cyclodipeptide oxidase is an enzyme filament, Andreas and Giessen present structural and functional analyses of the cyclodipeptide oxidase (CDO) AlbAB. To date, few CDOs have been characterized in depth, and all employ CDOA (nitroreductase-like) and CDOB (no known homology) subunits to perform O₂-dependent α,β -dehydrogenations on cyclodipeptides (CDPs). Diversification of CDPs is attractive, given their biologically relevant activities and their incorporation into various, more chemically complex drug molecules.

Through bioinformatics analyses presented in the manuscript, the team have considerably expanded the number of known CDOs, identified previously undescribed CDOA/CDOB fusions, and found numerous CDOs in non-CDPS-encoding gene clusters. Interestingly, they also observed CDO sequences in anaerobic bacteria, despite the expectation of O₂ as the terminal electron acceptor. In a significant stride forward with these previously intractable enzymes, the authors next expressed and purified AlbAB from the heterologous host *Streptomyces coelicolor*. Consistent with its appearance in the size-exclusion chromatography void volume, the protein was observed in long filaments by negative stain TEM analysis; these filaments were found to be stable under a range of pH values and ionic strengths.

The team also performed the first in vitro activity and kinetic analyses on a purified CDO, testing AlbAB against its native substrate cyclo(L-Phe-L-Leu) through an established two-enzyme reaction with horseradish peroxidase. Notably, AlbA could not be individually isolated, and AlbB was inactive on its own and failed to form filaments. However, AlbAB filaments retained their filamentous structure under reaction conditions, as shown by TEM. Formation of these filaments was further supported by size-exclusion chromatography and dynamic light scattering experiments.

Next, the authors carried out structure determination of AlbAB using single-particle cryo-EM. Here, they observed that dimers of each of AlbA and AlbB form a heterotetrameric unit along the helical filament. Moreover, from the cryo-EM density, FMN could be identified as the flavin cofactor, covalently linked to each AlbA monomer. The structures of AlbA and AlbB, their homodimer/heterodimeric interfaces, and FMN binding sites were subsequently discussed in detail. Finally, through agarose gel shift assays and size-exclusion chromatography, the authors demonstrated that neither the CDPS AlbC nor tRNA form a stable complex with AlbAB filaments.

As a whole, I find the manuscript clearly and elegantly written, the analysis comprehensive, and the results beautifully illustrated. The discovery and characterization of filament-forming enzymes involved in secondary metabolite

production is unexpected yet exciting in terms of biocatalytic potential, especially as CDPs are useful chemical precursors with important bioactivities. I suggest the submission to be highly suitable for publication in Nature Communications once the following points are addressed.

1. Can the structural results of the authors be used to better understand the substrate specificity and/or catalytic mechanism, and thus the potential of AlbAB for biocatalysis? For example, in silico docking (either manual or automated) of enzyme and substrate or product could yield insight into this aspect. Indeed, the authors have stated that “an α - β bond in a cyclic dipeptide substrate must be oriented above the N5 nitrogen of FMN to allow for hydride transfer”; they also suggested a catalytic role for Tyr37 of AlbB, which is reasonably given a similar role in homologous enzymes. These constraints should (hopefully) allow the substrate to be reasonably placed within the active site, especially as the amide NH and C=O atoms are likely to form specific H-bonds to nearby atoms.

Author reply: We thank Reviewer #1 for this suggestion and agree that it would be a useful addition to the manuscript. We have performed additional molecular docking simulations with AlbAB and its substrate cyclo(L-Phe-L-Leu). We have included a new figure panel highlighting these docking results to the manuscript (Supplementary Fig. 11a). We have further expanded the results section to include a discussion of these results:

Lines 358-366: “Molecular docking of cyclo(L-Phe-L-Leu) into the active site of AlbAB further supports this proposed mechanism, placing the substrate in orientations where the α or β carbons of cyclo(L-Phe-L-Leu) are located within 3 Å of the hydroxyl group of Y37 and within 4 Å of the N5 nitrogen of FMN (Supplementary Fig. 11a). Substrate orientations also suggest that the preference of AlbAB for hydrophobic residue-containing cyclic dipeptides may be based on the presence of an active site hydrophobic pocket formed by AlbA residues A58, L104, and I108, and the AlbB residue P34 (Supplementary Fig. 11b). However, more detailed structural studies will be needed to elucidate the details of AlbAB substrate preference.”

2. Preparing a few active site mutations and repeating activity assays would be useful in supporting the assignment of substrate-binding/catalytic residues. This could be particularly powerful when combined with docking results.

Author reply: We agree with Reviewer #1 and have created and characterized active site mutants for the two residues (S54 and Y37) suggested to be important for catalysis (mutants: S54A and Y37F). S54A showed a ca. 62-fold decrease in activity, while no activity could be detected for Y37F. This is consistent with their proposed importance. We have included the activity assay results for active site mutants S54A of AlbA and Y37F of AlbB in Supplementary Fig. 11c, as well as a discussion of these results. We have also provided an illustration of the proposed mechanism in Supplementary Fig. 11d.

Lines 366-374: “To further elucidate AlbAB catalysis, we created the active site mutants S54A (AlbA) and Y37F (AlbB) to test our hypothesized mechanism. Both mutations resulted in significantly reduced catalytic activities, with S54A resulting in an

approximately 62-fold decrease in catalytic activity, and Y37F exhibiting no measurable activity (Supplementary Fig. 11c). These results suggest a mechanism where S54 may interact with the Y37 side chain hydroxyl, allowing Y37 to function as the catalytic base responsible for proton abstraction from the α or β carbon of the cyclodipeptide substrate (Supplementary Fig. 11d). This proposed mechanism resembles the mechanisms suggested for some characterized NTR-like dehydrogenases...”

3. What does a DALI search (<http://ekhidna2.biocenter.helsinki.fi/dali/>) yield for AlbB? Are there closely related and/or interesting structures, despite the lack of sequence homology?

Author reply: We performed a DALI search for AlbB and found that it provided structural alignments for α -helical segments of various non-related proteins with little to no significant structural similarity to AlbB. We have added a discussion of these results:

Lines 250-255: “To further investigate if the α -helical assembly of AlbB may be found in other characterized proteins, we performed a DALI⁷⁷ search using a monomer of AlbB as a query. Aside from mostly fragments of α -helical bundles found in unrelated structures, the search provided no structural alignments that were significantly similar to AlbB, suggesting that AlbB represents a novel type of dimeric α -helical assembly.”

4. What is the rationale for testing activity at pH 4.1 (see line 155)?

Author reply: pH 4.1 corresponds to the predicted pI of AlbB, and we sought to observe if filament assembly was disrupted at this pH. We have modified the text to explain the rationale of this experiment more precisely:

Lines 156-158: “We further sought to identify a pH condition that disrupted filament assembly. We tested filament stability at pH 4.1, which corresponds to the pI of AlbB, and did not observe filaments suggesting that filament assembly is disrupted at low pH.”

5. What are the expected absorbance maxima for a non-covalently associated FMN molecule (see line 161)?

Author reply: Non-covalently bound FMN molecules in proteins generally have absorbance maxima around 360 and 450 nm corresponding to the FMN. However, these values can deviate between FMN-binding proteins based on the local environment surrounding the FMN binding site, making it difficult to determine covalent FMN binding solely by the UV absorbance maxima. We have added a further discussion of the absorbance properties of the bound FMN in AlbAB in the folded and unfolded state, as well as the absorbance properties of flavoproteins more generally. For further clarification of FMN absorbance, we have also updated Fig. 1g with a cleaner spectrum of folded and unfolded AlbAB:

Lines 161-172: “AlbAB is a flavoprotein with absorbance peak maxima at 343 nm and 448 nm, indicative of a bound flavin cofactor (Fig. 1g). These absorbance maxima are consistent with previously reported values for AlbAB of 343.5 nm and 447.5 nm⁵⁰. We

found that flavin absorbance was still present after AlbAB denaturation in 6 M guanidinium hydrochloride followed by washing steps, confirming that the flavin cofactor is covalently bound as previously reported⁵⁰. Additionally, we observed a shift in the flavin absorbance maxima to ca. 372 nm and 452 nm in denatured AlbAB. The flavin absorbance maxima in both folded and denatured AlbAB are consistent with other nitroreductases and flavin-binding proteins, most of which bind flavins non-covalently and generally exhibit absorbance maxima at ca. 360 nm and 450 nm, although some deviation in absorbance must be expected due to the local protein environment surrounding the flavin cofactor⁶⁹⁻⁷².”

6. Other corrections:

a. Line 147: Please change “equimolar” to “approximately equimolar”, as concentration estimation by SDS-PAGE is not very accurate.

Author reply: We have changed this line to include “approximately equimolar”.

b. Line 365: “elude” should be “elute”.

Author reply: We have corrected this typo.

c. Line 384: Something is missing here – possibly “and” in “...filaments, [and] no protein...”.

Author reply: We have added “and” to this sentence. It now reads “Negative stain TEM analysis of AlbAB-containing fractions from gel filtration runs did not show any apparent difference compared to freshly purified AlbAB filaments, and no protein or tRNA directly interacting with filaments could be observed (Fig. 6c).”

d. Line 566: “5,000 x G” should read “5,000 x g”.

Author reply: We have changed this to 5,000 x g.

e. Line 607: Please define the abbreviation “CTF”.

Author reply: We have changed this sentence to state that CTF is the abbreviation for contrast transfer function: “3,015 movies were motion corrected using Patch Motion Correction, followed by contrast transfer function (CTF) estimation using Patch CTF Estimation.”

f. Lines 623-624: Please provide literature references for the Phenix and Coot software packages.

Author reply: These references are now included in the manuscript.

g. Figure S3 caption: Please define 4-HPA.

Author reply: We have changed 4-HPA to “4-hydroxyphenylacetic acid” in the caption of Supplementary Figure 3.

Reviewer #2:

In this work, Andreas and Giessen present cryo-EM and other data supporting the AlbAB cyclodipeptide oxidase (CDO) as a heterooligomeric enzyme filament. They also provide bioinformatics analyses of a collection of approximately 275 other CDOs, suggesting similarities in structure and function of these CDOs with AlbAB. The results of this work should be of interest to the biosynthesis community. Overall, their data are compelling and the evidence is well presented. There are a few concerns as noted below:

—The flow of Introduction may be improved by breaking the first paragraph into two paragraphs, perhaps with a break point after the CDPS/RCDPS discussion.

Author reply: We agree with Reviewer #2 and have broken the first introductory paragraph into two paragraphs.

—Fig 1h lacks error bars and does not indicate replication of the kinetics assay. Adequate replicates should be incorporated.

Author reply: We thank Reviewer #2 for pointing out this oversight and have added error bars and individual replicate data points to Fig 1h.

—Under “AlbAB is a filament-forming CDO with covalently bound flavin cofactors” section, it is mentioned that purified AlbAB has a “very high apparent molecular weight.” This statement would be more impactful if a weight range were stated.

Author reply: We have clarified this statement:

Lines 148-151: “As reported previously⁵⁰, we observed the majority of AlbAB eluting at or near the void fractions when purified using a Superose 6 Increase 10/300 GL column, suggesting that the majority of AlbAB oligomers possess apparent molecular weights exceeding 2 megadaltons.”

—Paragraph 4 of Discussion: Since the experimental data of the paper focus on AlbAB rather than CDO-like proteins in non-CDPS clusters, much of this paragraph is speculative in nature and overly redundant of bioinformatics discussions presented in the Results. This paragraph should be made more focused and succinct.

Author reply: We agree with Reviewer #2 and have modified this paragraph to be less speculative. We maintain the statements describing some of the potential roles of non-CDPS-associated CDO-like proteins in modifying hydantoin- or allantoin-related molecules, as this is supported by conserved genes encoding allantoin and hydantoin modifying enzymes in many of the non-CDPS-associated CDO-like gene clusters. We

have further removed the discussion on the role of the covalently bound flavin to avoid speculation and suggest that further experimental validation is necessary to elucidate the roles of the non-CDPS-associated CDO-like proteins.

Lines 475-480: “We cannot exclude the possibility that non-CDPS-associated CDO-like enzymes may not function as oxidases or dehydrogenases. The broad substrate repertoire and various reactions performed by NTR-like enzymes makes it difficult to predict the biological functions of the non-CDPS-associated CDO-like enzymes. Further experimental investigation is necessary to elucidate the function of these enzymes.”

Reviewer #3:

The manuscript submitted by Andreas and Giessen focuses on the structural and functional characterization of AlbAB, a cyclodipeptide oxidase (CDO) from *Streptomyces noursei* involved in albonoursin biosynthesis. The authors report that AlbAB forms a megadalton heterooligomeric enzyme filament with covalently bound flavin mononucleotide cofactors, and they emphasize the importance of filament formation for enzyme activity. The study suggests that AlbA-AlbB interactions are conserved, indicating a potential generalization to all CDO-like enzymes functioning as enzyme filaments. The work addresses historical challenges in studying CDOs and provides insights into their structural and functional aspects. The authors anticipate implications for biocatalysis and chemoenzymatic synthesis. Overall, the manuscript contributes to the understanding of CDOs, marking a notable advancement in the field of enzymology and natural product biosynthesis.

In this work, single-particle cryo-electron microscopy (cryo-EM) played a pivotal role in unveiling the molecular structure and assembly of the AlbAB filament. The technique was specifically employed to provide detailed insights into the architecture of the cyclodipeptide oxidase (CDO) AlbAB, allowing the visualization of the arrangement of AlbA and AlbB subunits within the heterooligomeric enzyme filament. This detailed structural information was crucial for understanding the organization of the AlbAB filament, offering a foundation for future investigations into the functional mechanisms of CDOs and providing valuable insights into their catalytic activity and substrate interactions.

The authors have undertaken a helical reconstruction employing the single-particle analysis implementation within CryoSPARC, coupled with a local refinement approach. Their meticulous approach to the structural analysis reflects a technically rigorous examination. The resulting characterization demonstrates a thorough and detailed understanding of the structural intricacies, showcasing a commendable level of precision in their investigative methods.

The manuscript is well-crafted, demonstrating clarity in its writing style that enhances readability. Despite the intricacy of the analysis, the text remains reasonably approachable for individuals who may not be specialists in the field. Only a few minor changes are needed to further refine the document.

Minor Comments

Pp7 II 185-186: “Initial 2D class averages displayed clear periodicity highlighting two distinct subunits along the length of the filament (Fig. 2a)”

The presence of the two distinct subunits is not immediately apparent in the initial 2D class averages. To enhance clarity, especially for non-expert readers, it is recommended to highlight these subunits in at least one of the 2D averages.

Author reply: We thank Reviewer #3 for this suggestion and have highlighted the subunits in the 2D averages in Fig. 2a to clearly delineate AlbA and AlbB dimers within the 2D class averages.

Pp7 LI 183-198. Each AlbA₂ dimer and AlbB₂ dimer appears to exhibit a local 2-fold symmetry within the subunits that constitute the dimer. If confirmed, it should be described in the text.

Author reply: We have modified the text to now specifically mention the observed C₂ symmetry within the AlbA₂ and AlbB₂ dimers:

Lines 197-200: “3D helical reconstruction yielded a map with a global resolution of 3.14 Å where a single C₂ symmetrical dimer of AlbA (AlbA₂) and a single C₂ symmetrical dimer of AlbB (AlbB₂) form a heterotetramer representing the biologically relevant asymmetric unit of the filament (Fig. 2b).”

REVIEWERS' COMMENTS

Reviewer #1 (Remarks to the Author):

The authors have done an excellent job at addressing the points that I and other referees have raised, and I highly recommend their article be published in Nature Communications. I appreciate the inclusion of a proposed catalytic mechanism, but I think the illustration in Figure 11d could use clarification/correction. On the left side of the reaction arrow, the curved arrow showing hydride transfer should originate precisely from the C α -H α bond of the substrate; on right side of the reaction arrow, the authors should include the transferred hydride on N5 of FMN, redraw the N5=C4a and N10=C10a double bonds as single bonds while redrawing a new C4a=C10a double bond, and show a negative charge on N1, all as suggested by the curved arrows on the left side of the reaction arrow. For example, please have a look at Figure 7 panels II-III of Imagawa et al., 2011, J. Biol. Chem., <https://doi.org/10.1074/jbc.M111.257824>.

In another minor edit, I suggest a colon replace the existing comma in lines 37-39, as written below:

DKP formation can be catalyzed by three evolutionarily unrelated types of enzymes: nonribosomal peptide synthetases (NRPSs)¹⁴⁻²¹, cyclodipeptide synthases (CDPSs)²²⁻²⁸, and arginine-containing cyclodipeptide synthases (RCDPSs)²⁹.

Also, in the Figure 1 caption, the B, D, and E references should be lower case, like a, c, f, g, and h.

Reviewer #2 (Remarks to the Author):

The authors have done an excellent job of addressing the concerns I raised in their earlier version. With these changes now in place, this manuscript should make an interesting and important addition to the natural product and biosynthesis fields.

Reviewer #4 (Remarks to the Author):

In this manuscript, Andreas and Giessen describe the structure of a cyclodipeptide oxidase (CDO), AlbAB, which they reveal to be a megadalton heterooligomeric enzyme filament. This work is significant

because it marks the first structural characterization of a CDO, an enzyme with potential applications in biocatalysis and chemoenzymatic synthesis. The work thus advances our understanding of CDO mechanisms and opens new avenues in the production and modification of bioactive compounds. The methodology appears robust, with sufficient detail for reproducibility. The data analysis, interpretation, and conclusions are well-supported. The text is well-written, and many of the figures are stunning. I recommend only minor revisions.

Page 16, line 343. "Without substrate bound, the active site above the re face of FMN has an approximate volume of 1,340 Å³ (Fig. 5e,f and Supplementary Fig. 10)." I believe the authors mean to say that the volume is 1,340 Å³ when the substrate is deleted from the holo structure. It could be that in the actual apo structure, this pocket would collapse. This is a minor point requiring only slight rewording to fix the ambiguity. Also, this section appears to be missing a citation. How was the volume measured?

Page 16, paragraph starting at line 351. I recommend breaking this very long paragraph into multiple paragraphs.

Page 29, line 720 (section "Molecular Docking"). I assume the authors used Webina to add protein/ligand hydrogen atoms appropriate for a given pH, but they might mention that explicitly. Webina is based on the Vina codebase. Per <https://vina.scripps.edu/manual/>, both the ligand and receptor must be appropriately protonated (i.e., "your ligand or receptor might not have been correctly protonated" is listed as a possible cause of poorly docked poses). Most readers will not realize that Webina differs from Vina in that it optionally handles the protonation step internally, so users do not need to perform this step separately. (I myself forgot that Webina handles protonation until I checked the website.)

Point-by-point response to reviewer comments:

Author replies shown in blue

Reviewer #1 (Remarks to the Author):

The authors have done an excellent job at addressing the points that I and other referees have raised, and I highly recommend their article be published in Nature Communications. I appreciate the inclusion of a proposed catalytic mechanism, but I think the illustration in Figure 11d could use clarification/correction. On the left side of the reaction arrow, the curved arrow showing hydride transfer should originate precisely from the C α -H α bond of the substrate; on right side of the reaction arrow, the authors should include the transferred hydride on N5 of FMN, redraw the N5=C4a and N10=C10a double bonds as single bonds while redrawing a new C4a=C10a double bond, and show a negative charge on N1, all as suggested by the curved arrows on the left side of the reaction arrow. For example, please have a look at Figure 7 panels II-III of Imagawa et al., 2011, J. Biol. Chem., <https://doi.org/10.1074/jbc.M111.257824>.

In another minor edit, I suggest a colon replace the existing comma in lines 37-39, as written below:

DKP formation can be catalyzed by three evolutionarily unrelated types of enzymes: nonribosomal peptide synthetases (NRPSs)¹⁴⁻²¹, cyclodipeptide synthases (CDPSs)²²⁻²⁸, and arginine-containing cyclodipeptide synthases (RCDPSs)²⁹.

Also, in the Figure 1 caption, the B, D, and E references should be lower case, like a, c, f, g, and h.

Author reply: We thank Reviewer #1 for the comments. We have addressed all points raised by Reviewer #1. We have updated the mechanism in Supplementary Fig. 11d as requested and have fixed the highlighted typos.

Reviewer #2 (Remarks to the Author):

The authors have done an excellent job of addressing the concerns I raised in their earlier version. With these changes now in place, this manuscript should make an interesting and important addition to the natural product and biosynthesis fields.

Author reply: We thank Reviewer #2 for the kind comment.

Reviewer #4 (Remarks to the Author):

In this manuscript, Andreas and Giessen describe the structure of a cyclodipeptide oxidase (CDO), AlbAB, which they reveal to be a megadalton heterooligomeric enzyme filament. This work is significant because it marks the first structural characterization of a CDO, an enzyme with potential applications in biocatalysis and chemoenzymatic synthesis. The work thus advances our understanding of CDO mechanisms and opens new avenues in the production and modification of bioactive compounds. The methodology appears robust, with sufficient detail for reproducibility. The data analysis, interpretation, and conclusions are well-supported. The text is well-written, and many of the figures are stunning. I recommend only minor revisions.

Page 16, line 343. "Without substrate bound, the active site above the re face of FMN has an

approximate volume of 1,340 Å³ (Fig. 5e,f and Supplementary Fig. 10).” I believe the authors mean to say that the volume is 1,340 Å³ when the substrate is deleted from the holo structure. It could be that in the actual apo structure, this pocket would collapse. This is a minor point requiring only slight rewording to fix the ambiguity. Also, this section appears to be missing a citation. How was the volume measured?

Author reply: We believe Reviewer #4 misinterpreted this sentence. Our original statement is correct as our structure is the apo structure of the enzyme filament, i.e. with no cyclic dipeptide bound. It is this experimentally determined structure upon which our calculation is based. We do reference the tool used to calculate the volume in the caption of Supplementary Fig. 10 (FPocketWeb V1.0.1).

Page 16, paragraph starting at line 351. I recommend breaking this very long paragraph into multiple paragraphs.

Author reply: We agree and have added additional paragraph breaks.

Page 29, line 720 (section “Molecular Docking”). I assume the authors used Webina to add protein/ligand hydrogen atoms appropriate for a given pH, but they might mention that explicitly. Webina is based on the Vina codebase. Per <https://vina.scripps.edu/manual/>, both the ligand and receptor must be appropriately protonated (i.e., “your ligand or receptor might not have been correctly protonated” is listed as a possible cause of poorly docked poses). Most readers will not realize that Webina differs from Vina in that it optionally handles the protonation step internally, so users do not need to perform this step separately. (I myself forgot that Webina handles protonation until I checked the website.)

Author reply: We thank Reviewer #4 for pointing this out. We have specifically added a sentence to the “Molecular Docking” section that now states that appropriate ligand and receptor protonation at pH 7.4 was automatically carried out as part of the Webina workflow.